# Multi-Task GRPO: Reliable LLM Reasoning Across Tasks

**Shyam Sundhar Ramesh** [1 2]  **Xiaotong Ji** [3]  **Matthieu Zimmer** [3]  **Sangwoong Yoon** [4]  **Zhiyong Wang** [5]
**Haitham Bou Ammar** [2 3]  **Aurelien Lucchi** [† 6]  **Ilija Bogunovic** [† 2 6]

## Abstract

RL-based post-training with GRPO is widely used to improve large language models on individual reasoning tasks. However, real-world deployment requires reliable performance across diverse tasks. A straightforward multi-task adaptation of GRPO often leads to imbalanced outcomes, with some tasks dominating optimization while others stagnate. Moreover, tasks can vary widely in how frequently prompts yield zero advantages (and thus zero gradients), which further distorts their effective contribution to the optimization signal. To address these issues, we propose a novel Multi-Task GRPO (MT-GRPO) algorithm that (i) dynamically adapts task weights to explicitly optimize worst-task performance and promote balanced progress across tasks, and (ii) introduces a ratio-preserving sampler to ensure task-wise policy gradients reflect the adapted weights. Experiments on both 3-task and 9-task settings show that MT-GRPO consistently outperforms baselines in worst-task accuracy. In particular, MT-GRPO achieves 16–28% and 6% absolute improvement on worst-task performance over standard GRPO and DAPO, respectively, while maintaining competitive average accuracy. Moreover, MT-GRPO requires 50% fewer training steps to reach 50% worst-task accuracy in the 3-task setting, demonstrating substantially improved efficiency in achieving reliable performance across tasks.

## 1. Introduction

Recent advances in RL post-training using policy optimization methods such as Group-Relative Policy Optimization (GRPO) have produced LLMs with impressive performance

on individual reasoning benchmarks, including mathematical problem solving, code generation, and structured reasoning tasks (Shao et al., 2024; Guo et al., 2025; Yu et al., 2025). Despite these advances, most post-training pipelines are designed and tuned primarily for individual tasks or benchmarks, treating each as an *isolated optimization target*, with limited work addressing cross-benchmark trade-offs or developing principled approaches for multi-task RL post-training. This becomes increasingly problematic as LLMs are deployed in real-world as general-purpose reasoners rather than specialists for narrow benchmarks, wherein broad competence across diverse reasoning skills is essential for reliability. A model that excels at competition mathematics but struggles with basic logical inference remains unreliable despite strong benchmark performance. This raises a fundamental question that current work largely sidesteps: *How should we post-train a single model to improve reasoning across tasks while ensuring that no task is left behind?*

Addressing this challenge is non-trivial. Standard multi-task post-training that optimizes for average performance often leads to imbalanced outcomes (Chen et al., 2025b; Akter et al., 2025), where strong gains on some tasks mask stagnation on others as illustrated in Figure 1. Moreover, joint post-training of tasks can introduce negative transfer and task interference, where progress on certain tasks hinders learning on others (Wu et al., 2020; Yu et al., 2020). Together, these issues highlight the need for *principled, robustness-aware optimization strategies* for multi-task post-training.

In this work, we incorporate task-wise robustness directly into the multi-task RL post-training objective to promote balanced competence across tasks. To make this objective operational in modern pipelines, we propose **MULTI-TASK GRPO (MT-GRPO)**, a novel post-training algorithm with two key ideas. First, it proposes *improvement-aware task reweighting*, using both task-level rewards and task-wise improvement signals to improve worst-task performance and overall multi-task robustness without sacrificing the average performance. Second, it introduces a *ratio-preserving, acceptance-aware batch construction mechanism* that enforces target task proportions in the training batch, ensuring that learned task weights translate into actual gradient signals (see Figure 1 for illustration).

---
[†] Co-senior authors. [1]UCL EEE [2]UCL Centre for AI [3]Huawei Noah's Ark Lab [4]UNIST Graduate School of AI [5]University of Edinburgh [6]University of Basel. Correspondence to: Shyam Sundhar Ramesh <shyam.ramesh.22@ucl.ac.uk>.

*Proceedings of the 43rd International Conference on Machine Learning*, Seoul, South Korea. PMLR 306, 2026. Copyright 2026 by the author(s).

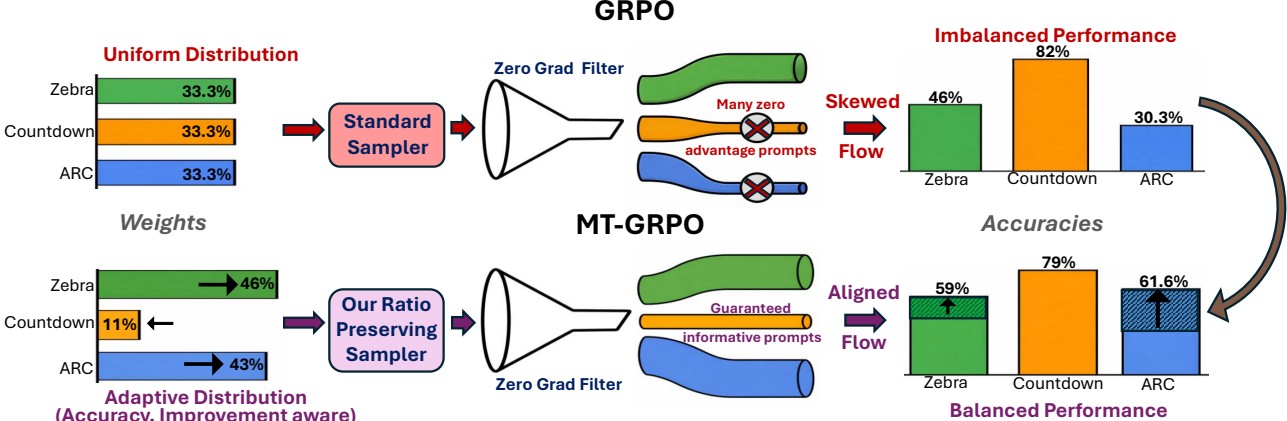

*Figure 1.* GRPO assigns uniform task weights and samples without regard to task difficulty or zero-gradient rates. Consequently, easy tasks (Countdown) dominate while harder tasks (ARC, Zebra) lag, and effective gradient flow is skewed by varying zero-gradient rates ($\otimes$ marks high zero-gradient rates). In contrast, MT-GRPO adapts task weights to prioritize weaker tasks and uses a ratio-preserving sampler to align effective gradient contributions with target weights, substantially improving ARC and Zebra and yielding more balanced performance.

Similar robustness-aware objectives have been studied in other learning paradigms, including distributionally robust optimization and multi-task learning (Namkoong & Duchi, 2016; Sagawa et al., 2019; Désidéri, 2012; Yu et al., 2020; Liu et al., 2023), as well as domain reweighting for large-scale pre-training (Oren et al., 2019; Xie et al., 2023; Liu et al., 2024b; Grangier et al., 2024; Diao et al., 2025). These methods typically operate by adapting weights over tasks or data groups based on their losses to optimize a target objective. However, in the context of RL-based LLM post-training with GRPO, prompts whose rollouts receive identical rewards yield zero advantages and contribute no gradient to the policy update. Since the prevalence of such prompts varies across tasks, effective gradient contributions are dominated by tasks with more non-zero gradient prompts, even when weaker tasks are intentionally upweighted (see Figure 1). Moreover, the GRPO loss is unreliable for task reweighting since it takes similar values when a prompt's rollouts are all correct or all incorrect. These challenges do not arise in prior settings and require algorithmic solutions beyond existing robust optimization techniques.

We detail other related works extensively in Section A and summarize our **main contributions** below.

(i) A robustness-aware multi-task RL post-training objective with a tunable trade-off between worst-task robustness and average performance.

(ii) A task reweighting framework that leverages task-level rewards and task-wise improvements to encourage balanced progress across tasks.

(iii) A ratio-preserving batch construction that aligns task weights with realized gradient contributions.

(iv) Empirical evaluation of MT-GRPO by post-training 3B and 7B models on multi-task reasoning benchmarks spanning planning (Countdown, Zebra puzzles), inductive reasoning (ARC), natural language QA (SciKnow-Eval), and mathematics, across controlled and larger multi-task settings.

Across settings, we observe that MT-GRPO improves worst-task accuracy over strong baselines while maintaining competitive average performance, and reallocates optimization effort toward weaker or slowly improving tasks.

## 2. Problem Formulation

We consider a collection of $K$ reasoning tasks indexed by $k \in [K] := \{1, 2, \ldots, K\}$, where each task $k$ is associated with a dataset $D_k$ and a reward function $R_k$. The datasets $D_k$ consist of a disjoint set of prompts, where each prompt contains a question specific to task $k$. The questions admit verifiable correct answers, and the reward function $R_k$ is designed to evaluate both the correctness and formatting of responses to questions from task $k$. Let $\pi_\theta$ denote a base policy with general reasoning capabilities. Our objective is to post-train $\pi_\theta$ jointly over these $K$ tasks.

For each task $k$, we define the task-level performance metric $J_k(\theta)$, which denotes the KL-regularized expected reward attained by policy $\pi_\theta$ on questions sampled from $D_k$. The KL regularizer explicitly penalizes deviations from a reference policy $\pi_{\text{ref}}$, ensuring that post-training does not excessively alter the model's behavior. Concretely, we define

$$J_k(\theta) := \mathbb{E}_{\substack{x \sim D_k \\ y \sim \pi_\theta(\cdot|x)}} \left[ R_k(x, y) - \tau \, \text{KL}\left(\pi_\theta \| \pi_{\text{ref}}\right) \right], \quad (1)$$

where $\tau$ controls the strength of KL regularization. A standard RL-based post-training approach for optimizing the policy $\pi_\theta$ over $K$ tasks is to maximize the average KL-regularized expected reward,

$$\max_{\theta \in \Theta} J_{\text{avg}}(\theta) = \frac{1}{K} \sum_{k=1}^{K} J_k(\theta). \quad (2)$$

**Policy gradient algorithms:** For the single-task case, the objective in Equation (2) is commonly optimized using policy gradient algorithms such as RLOO, PPO, GRPO, VinePPO, etc., (Ahmadian et al., 2024; Schulman et al., 2017; Shao et al., 2024; Kazemnejad et al., 2024). They apply the policy gradient theorem (Sutton et al., 1999),

$$\nabla_\theta J(\theta) = \mathbb{E}_{x,y}\Big[\nabla_\theta \log \pi_\theta(y \mid x)\big(\tilde{A}(x,y)\big)\Big],$$

where $\tilde{A}(x,y) = A(x,y) - \tau \log \frac{\pi_\theta(y|x)}{\pi_{\text{ref}}(y|x)}$ and $A(x,y)$ denotes an advantage function measuring the relative quality of response $y$ for prompt $x$ in comparison to the current behavior of the policy. Among these approaches, GRPO has recently emerged as a particularly effective and widely adopted method for improving the reasoning capabilities of large language models (Shao et al., 2024; Guo et al., 2025; Aggarwal & Welleck, 2025; Hu et al., 2025). It avoids the need for a value function and leverages within-prompt relative comparisons to construct stable advantage estimates. In particular, GRPO constructs a prompt-level, sample-based advantage. For each prompt $x$, we sample a group of $G$ responses $\{y_i\}_{i=1}^G$ from a behavior policy $\pi_{\theta_{\text{old}}}$ and define a relative advantage via within-group normalization, e.g., $A(x,y_i) = \Big(R(x,y_i) - \text{mean}(\{R(x,y_j)\}_{j=1}^G)\Big)/\text{std}(\{R(x,y_j)\}_{j=1}^G)$.

Using importance weighting to correct for off-policy sampling, the resulting GRPO objective is

$$J_{\text{GRPO}}(\theta) = \mathbb{E}_x\Big[\mathbb{E}_{\{y_i\}\sim\pi_{\theta_{\text{old}}}}\Big[\frac{1}{G}\sum_{i=1}^G \frac{\pi_\theta(y_i \mid x)}{\pi_{\theta_{\text{old}}}(y_i \mid x)} \\ \cdot \Big(A(x,y_i) - \tau \log \frac{\pi_\theta(y_i|x)}{\pi_{\text{ref}}(y_i|x)}\Big)\Big]\Big]. \quad (3)$$

For task $k$, $J_{\text{GRPO},k}(\theta)$ denotes the GRPO objective restricted to prompts $x \sim D_k$.

In practice, GRPO employs a clipped version of Equation (3), where the importance ratio $\rho_i = \pi_\theta(y_i \mid x)/\pi_{\theta_{\text{old}}}(y_i \mid x)$ is clipped to the interval $[1-\epsilon, 1+\epsilon]$. This prevents excessively large updates caused by samples with high importance weights and improves training stability. For brevity, we provide the clipped version in Section B.

**Limitations of standard GRPO in the multitask setting:** Despite its advantages, directly applying GRPO to the multitask objective in Equation (2) by averaging losses across tasks leads to two important issues.

**(i) Lack of task-wise robustness:** Optimizing average reward is fundamentally misaligned with the goal of general-purpose reasoning. The mean objective permits solutions in which strong gains on a subset of tasks compensate for substantial underperformance on others, providing no guar-

antees of task-wise robustness. This often leads to imbalanced outcomes in heterogeneous collections of reasoning tasks (Chen et al., 2025b; Akter et al., 2025). As a result, models post-trained to maximize average performance may lack balanced competence across diverse reasoning skills.

**(ii) Uneven zero-gradient rates across tasks:** A structural limitation of extending GRPO to the multi-task setting is that different tasks exhibit widely different rates of zero-gradient prompts. Under GRPO, when all sampled rollouts for a prompt receive identical rewards, the resulting advantages are zero and the prompt contributes no gradient to the policy update (Yu et al., 2025; Foster et al., 2025). Yu et al. (2025) addresses this issue in the single-task setting by filtering such prompts and resampling. However, this mechanism is insufficient for multi-task training, as post-filtered batches would become biased toward tasks with fewer zero-gradient prompts, which may even hurt overall average performance. More importantly, when weaker tasks are explicitly upweighted, this causes the realized gradient contributions to deviate substantially from the intended task proportions and leaves them under-optimized.

These challenges motivate the need for modified objectives and optimization strategies that preserve the practical benefits of GRPO while emphasizing task-wise robustness and ensuring appropriate gradient contributions from each task.

## 3. Multi-Task Post-Training Objective

In this section, we primarily tackle the task-wise robustness issue in the standard average reward RL objective (see Equation (2) and Limitation (i) in Section 2). We note that this involves designing a novel multi-task post-training objective that explicitly controls performance disparities across tasks and balances robustness and average performance. Our goal is to improve task-level rewards while ensuring *robust performance across tasks*. In particular, we seek a post-trained policy that satisfies the following desiderata:

(i) **High average performance:** The average rewards across all tasks is maximized.

(ii) **Robust across tasks:** The difference in rewards between any two tasks is bounded, ensuring that no task significantly underperforms relative to others.

Together, these criteria promote a post-trained policy that achieves balanced competence across different reasoning tasks. We formalize these two objectives using the following constrained optimization problem:

$$\max_{\theta\in\Theta} \quad \frac{1}{K}\sum_{k=1}^K J_k(\theta) \quad \text{s.t.} \quad |J_k(\theta) - J_j(\theta)| \leq \varepsilon, \quad (4)$$

where $\forall\, 1 \leq k < j \leq K, \varepsilon \geq 0$ controls the allowable performance disparity between tasks. Setting $\varepsilon = 0$ enforces

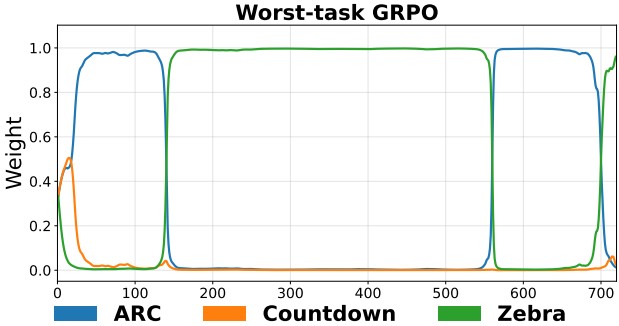

*Figure 2.* In strict worst-task optimization ($\varepsilon = 0$), task weights rapidly collapse to the current worst task and oscillate as the worst task shifts, resulting in near-zero weighting of Countdown.

strict equality of task performance, while larger values progressively relax the constraint toward average reward optimization. The constraints explicitly encourage minimizing disparities across tasks rather than allowing large gains on some tasks to compensate for only marginal gains on others.

For tractability, we work with the max–min objective obtained via a Lagrangian reformulation of Equation (4) (see Section C.1 for the derivation):

$$\max_{\theta \in \Theta} \min_{z \in \Delta_K} \sum_{k=1}^{K} z_k J_k(\theta) + \varepsilon\, \Omega(z), \qquad (5)$$

where $z \in \Delta_K$ denotes a learned distribution over tasks, and the regularizer $\Omega(z) = \frac{1}{2}\left\| z - \frac{1}{K}\mathbf{1} \right\|_1$ penalizes deviations from uniform weighting. Next, we develop an RL-based post-training algorithm for optimizing Equation (5) and analyzes its implications for task-wise robustness.

### 3.1. Adapting GRPO for Worst-Task Reward Maximization

We begin by considering the case $\varepsilon = 0$, which enforces the strongest notion of task-wise robustness and yields a minimax objective for multi-task post-training. This setting has been extensively studied in the literature as distributionally robust optimization (Sagawa et al., 2019; Oren et al., 2019; Xie et al., 2023). A common theme in such approaches to minimax or group-robust objectives is to alternate between updating the model parameters and updating group weights $z$, where groups with higher loss are assigned larger weights. This inherently assumes that the loss is a reliable scalar signal that reflects the group's performance.

In the multi-task GRPO setting, however, this assumption no longer holds. The GRPO objective is based on probability-weighted advantages and can evaluate to zero both when all sampled responses are incorrect and when all sampled responses are correct. While this behavior is acceptable for updating the policy parameters $\theta$, it introduces ambiguity when comparing across tasks for updating weights: a task on which the policy completely fails can appear indistinguishable (in terms of $J_{\mathrm{GRPO}}$) from a task on which the

policy performs perfectly. In a multi-task setting, where task weights must be adapted based on task performance, such ambiguity can lead to systematically misleading updates. To our knowledge, this issue has not been discussed in prior works, as it arises specifically from the structure of GRPO-style objectives used in modern LLM post-training.

To address this, our idea is to decouple task reweighting from the GRPO loss and instead use true task-level rewards $J_k(\theta)$ to update task weights. This design choice is specific to the post-training + GRPO setting and constitutes an important departure from existing robust learning methods.

**Update rule:** Given this observation and the subsequent design choice, to update task weights $z \in \Delta_K$, we define $z$ as $z_t = \mathrm{Softmax}(\xi_t)$ over logits $\xi \in \mathbb{R}^K$, and update $\xi$ instead. This allows unconstrained optimization over $\xi$ while ensuring $z_t \in \Delta_K$. At iteration $t$, we update $\xi_t$ through gradient descent w.r.t. weighted task rewards $L(\xi_t) = \sum_{k=1}^{K} z_{k,t} J_k(\theta_t)$ for fixed $\theta_t$. The resulting task-wise gradient, $(g_t)_k = z_{k,t}\Big(J_k(\theta_t) - \sum_{j=1}^{K} z_{j,t} J_j(\theta_t)\Big)$, is negative for tasks whose rewards fall below the current weighted average, thereby increases their corresponding logits and weights. This yields the following alternating updates:

$$\theta_{t+1} = \theta_t + \gamma_t \sum_{k=1}^{K} z_k^t \nabla_\theta J_{\mathrm{GRPO},k}(\theta_t), \qquad (6)$$

$$\xi_{t+1} = \xi_t - \beta\, g_t, \quad g_t = \begin{bmatrix} \nabla_\xi z_{1,t}^\top \\ \vdots \\ \nabla_\xi z_{K,t}^\top \end{bmatrix} \begin{bmatrix} J_1(\theta_t) \\ \vdots \\ J_K(\theta_t) \end{bmatrix}. \qquad (7)$$

These updates train the policy using a $z$-weighted GRPO loss across the $K$ tasks while adaptively adjusting the weights $z$ to prioritize underperforming ones, thereby encouraging more balanced performance across tasks.

**Issues with strict worst-task reward maximization:** While the alternating updates in Equation (6) correctly optimize the minimax objective in Equation (5) for $\varepsilon = 0$, they can lead to degenerate dynamics in which training is dominated by a single worst-performing task. This behavior is inherent to the inner problem in Equation (5) as for fixed $\theta$, the inner optimization over weights $z \in \Delta_K$ places all mass on the lowest-reward task for $\varepsilon = 0$.

With the softmax parameterization $z = \mathrm{Softmax}(\xi)$ and gradient descent on $\xi$, this tendency is further amplified by the exponential mapping. Tasks with lower rewards are repeatedly upweighted, causing $z$ to rapidly collapse toward a near one-hot distribution that persists until another task becomes worse. This behavior is visible in Figure 2, where the weight assigned to the current worst task quickly spikes toward one, while the other tasks receive near-zero weight for extended periods (e.g., Countdown is almost entirely

---

**Subroutine 1** Improvement-aware Weight Update (**IWU**)

1: **Input:** rewards $\{J_k(\theta_t)\}_{k=1}^K$, improvements $\{I_k^{(t)}\}_{k=1}^K$, logits $\xi_t$, stepsize $\beta$, trade-off $\lambda$
2: $z_t \leftarrow \text{Softmax}(\xi_t)$
3: $s_k^{(t)} \leftarrow I_k^{(t)} + \lambda J_k(\theta_t) \;\; \forall k \in [K]$
4: $(g_t)_k \leftarrow z_{k,t}\big(s_k^{(t)} - \sum_{j=1}^K z_{j,t} s_j^{(t)}\big) \;\; \forall k \in [K]$
5: $\xi_{t+1} \leftarrow \xi_t - \beta \, g_t$
6: **Return:** $z_{t+1} = \text{Softmax}(\xi_{t+1})$

---

ignored after the early steps). As a result, non-worst tasks are systematically under-optimized.

**3.2. Improvement-Aware Task Reweighting**

Motivated by the above limitation, we consider two mechanisms to stabilize task reweighting: (i) incorporation of task-level improvement, and (ii) $l_2$ regularization. We focus on (i) in the main text and analyze (ii) in Section F.

Absolute reward does not distinguish between tasks that are improving rapidly and tasks that have stagnated during training. A task with low reward but strong improvement may require less prioritization than a task with similar reward that no longer benefits from updates. As a result, reward-based reweighting alone can leave some tasks under-optimized. This motivates tracking how each task's loss (i.e., $-J_{\text{GRPO},k}(\theta)$) evolves over training and introducing an improvement-aware signal that captures how much each task benefits from policy updates.

**Task-level improvement:** Building on the notion of task-level improvement introduced in prior work Liu et al. (2023), we define the per-step improvement of task $k$ for multi-task post-training using GRPO as

$$I_k^{(t)} := J_{\text{GRPO},k}(\theta_{t+1}) - J_{\text{GRPO},k}(\theta_t). \quad (8)$$

This quantity captures whether the task-wise GRPO loss is improving, stagnating, or degrading. Moreover, it provides a measure of how much each task benefits from the policy update $\theta_{t+1} = \theta_t + \gamma_t \sum_{k=1}^K \nabla_\theta J_{\text{GRPO},k}(\theta_t)$.

**Improvement-aware reweighting:** Our goal is to prioritize tasks that are both underperforming in terms of $J_k(\theta_t)$ and under-improving in terms of $I_k^{(t)}$. We therefore update task-weight logits using the combined signal $I_k^{(t)} + \lambda J_k(\theta_t)$, where $\lambda$ controls the trade-off between task reward and task improvement. For large $\lambda$, the update approaches strict worst-task reward maximization, and for small $\lambda$, it promotes balanced improvement across tasks. We present these improvement-aware updates in Subroutine 1 (see Lines 4, 5).

Intuitively, Subroutine 1 accounts for task stagnation and deterioration when updating weights, rather than repeatedly upweighting tasks with lower rewards. This prevents collapse onto a single task and promotes balanced progress across tasks (see Section C.3 for a formal analysis).

---

**Algorithm 1** MULTI-TASK GRPO (MT-GRPO)

1: **Input:** $\mathcal{D}_{\text{train}} = \{D_1, \ldots, D_K\}$, batch size $B$, rollouts per prompt $N$, reward objective $J(\cdot)$, initial policy parameters $\theta^0$, initial filtered ratios $\rho^0$, initial task-weight logits $\xi^0$ ($z^0 = \text{Softmax}(\xi^0)$), total steps $T$
2: **for** $t = 0$ to $T-1$ **do**
3:     **Batch construction:**
    $(x, \rho^{t+1}) \sim \text{RP SAMPLER}\,(z^t, B, N, \rho^t, \mathcal{D}_{\text{train}})$
4:     **Standard GRPO update w.r.t. batch $x$:**
    $\theta^{t+1} \leftarrow \text{OPTIMIZER}\big(\theta^t, \nabla_\theta J_{\text{GRPO}}(\theta^t; x)\big)$
5:     $I_k^{(t)} \leftarrow J_{\text{GRPO},k}(\theta_{t+1}) - J_{\text{GRPO},k}(\theta_t) \;\; \forall k \in [K]$
6:     $z_{t+1} = \text{\textbf{IWU}}(\xi_t, z_t, I^{(t)}, J(\theta_t))$ (Subroutine 1)
7: **end for**
8: **Return** final policy parameters $\theta^T$

---

## 4. Algorithm

We present MULTI-TASK GRPO (MT-GRPO), a novel post-training algorithm for improving reasoning across multiple tasks. Our method addresses key limitations (see Section 2) that arise when adapting GRPO-style RL post-training to the multi-task setting, and is summarized in Algorithm 1. MT-GRPO jointly learns (i) the policy parameters and (ii) a distribution over tasks that governs how prompts are sampled during training. This distribution is dynamically updated to balance robustness and avg. performance (Limitation (i)), guided by task-level rewards and task-wise improvement signals (IWU in Subroutine 1). Moreover, MT-GRPO ensures consistency between learned task weights and effective gradient contributions (Limitation (ii)), by using RATIO-PRESERVING SAMPLER (RP SAMPLER) for batch construction.

**Adaptive task reweighting (IWU):** At each step, MT-GRPO updates task weights $z^t \in \Delta_K$ that control how prompts are sampled across tasks. We update these weights using a $\lambda$ weighted combination of task reward $J_k(\theta_t)$ and task improvement $I_k^{(t)}$ (Subroutine 1), where $\lambda$ plays a role analogous to $\varepsilon$ in Equation (5): larger $\lambda$ emphasizes worst-task robustness, while smaller $\lambda$ favors average performance. This prioritizes tasks that are underperforming or improving slowly by increasing their sampling frequency in subsequent batches. The improvement-aware update also prevents weight collapse onto a single worst task and the consequent under-optimization of other tasks. As a result, MT-GRPO achieves strong worst-task without sacrificing average performance, consistent with our objective in Equation (5).

**Uneven zero-gradient rates (RP SAMPLER):** In practice, sampling prompts according to task weights $z^t$ is insufficient under GRPO because many prompts yield zero gradients. Since tasks might exhibit widely different zero-gradient rates, the effective composition of the training batch can deviate from the intended task proportions, leading to a mismatch between the task weights produced

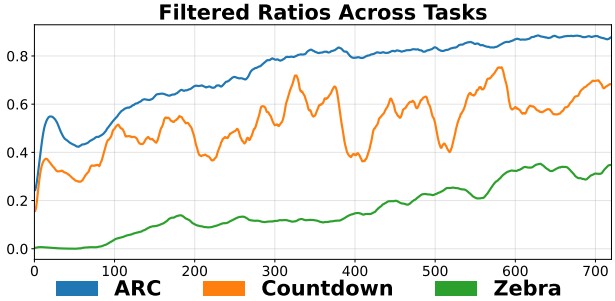

**Figure 3.** Ratios of zero-gradient prompts across tasks during training. ARC exhibits a much higher ratio than Zebra.

by Subroutine 1 and the actual gradient contributions. To address this, MT-GRPO uses a *Ratio-Preserving Sampler (RP sampler)* for batch construction (Algorithm 2). The RP sampler enforces the target task proportions in the post-filtered batch (after zero gradient prompts filtered) using oversampling and acceptance-aware resampling. We detail this procedure in Section 5.

Together, task weights influence how data are sampled, and the resulting reward and improvement metrics affect subsequent weight updates forming an effective loop that results in MT-GRPO's reliable performance across tasks.

## 5. Practical Findings and Solutions

While the task-weight updates in Subroutine 1 provide a principled mechanism for balancing performance across tasks, their efficacy depends on whether these weights translate into actual gradient contributions during training (see limitation (ii) in Section 2). In this section, we discuss the underlying mechanism in RP SAMPLER (Algorithm 2), a key component of Algorithm 1, that addresses this issue and ensures faithful realization of the intended task weights.

**Uneven Zero Gradient Rates:** A structural limitation of GRPO is that when all sampled responses for a prompt receive identical rewards, the resulting advantages are zero and the gradient vanishes for that prompt (Yu et al., 2025; Foster et al., 2025). The prevalence of such zero-gradient samples varies substantially across tasks (see Figure 3). Consequently, in a multi-task setting, even if two tasks are assigned equal weight $z_k$, the task producing informative gradients more frequently will contribute disproportionately to parameter updates. This systematically skews the training mixture away from the intended task proportions.

**Fix: Enforcing target task ratios** To correct this mismatch, we explicitly enforce task proportions (in the post-filtered batch) *after* filtering out zero-gradient prompts. Let $z \in \Delta_K$ denote task weights from Subroutine 2 and $B$ the target batch size. We first sample desired post-filtered counts via $(n_1, \ldots, n_K) \sim \text{Multinomial}(B, z)$, where $n_k$ specifies the number of informative samples from task $k$ in the final

batch. We then generate and filter samples, tracking non-zero-gradient samples $c_k$ per task. We resample prompts until $c_k \geq n_k$ or a fixed regeneration budget is exhausted.

**Inefficiency Under High Filtering Rates:** While the above procedure ensures correctness, it can be inefficient when some tasks exhibit high filtering rates, requiring many regeneration rounds to meet post-filtered targets.

**Fix: Acceptance-aware sampling.** To reduce regeneration overhead, we introduce an acceptance-aware sampling strategy that anticipates task-dependent filtering. We maintain an estimate $\rho_k$ of the filtering rate (fraction of generated samples with zero gradients) for each task $k$. During sampling, we inflate task weights as $\hat{z}_k = \frac{z_k m_k}{\sum_{j=1}^{K} z_j m_j}$, $\quad m_k = \min\left\{\frac{1}{1-\rho_k}, M_{acc}\right\}$, where $M_{acc}$ caps the inflation factor. Tasks with higher expected filtering are oversampled during generation, increasing the likelihood that the post-filtered batch matches the desired proportions. During resampling, we similarly prioritize tasks based on their deficiency ($c_k$ relative to $n_k$) and expected acceptance. This strategy reduces regeneration overhead while preserving consistency with the task proportions induced by Subroutine 1.

## 6. Experiments

We evaluate MT-GRPO (Algorithm 1) in a multi-task RL post-training setting on reasoning tasks spanning planning and inductive reasoning. We post-train the Qwen-2.5-3B base model on three task families from Chen et al. (2025b).

**Planning:** (i) *Countdown*: Given 3–5 integers, the model applies arithmetic operations to reach a target value. (ii) *Zebra puzzles*: Logic puzzles over 3–5 entities and properties with textual constraints to infer the correct assignment. In both cases, difficulty increases with number of input values.

**Inductive reasoning:** *Abstraction and Reasoning Corpus (ARC)*: Each instance contains 3 input–output examples illustrating a transformation rule, and the model must generalize to a test example. We use string-based ARC tasks of lengths 10, 20, and 30, with length determining difficulty.

**Datasets.** We use datasets released by Chen et al. (2025b) generated via Stojanovski et al. (2025). Each task family includes three difficulty levels (easy, medium, hard), with 10k training instances and 200 evaluation instances per level.

**Baselines:** We compare against four competitive baselines: **(i) GRPO:** Uniform sampling over tasks when constructing training batches. **(ii) SEC-GRPO:** Self-evolving curriculum (Chen et al., 2025b) prioritizing tasks with larger absolute advantages. **(iii) DAPO:** Uniform sampling with DAPO-style clipping and dynamic sampling (Yu et al., 2025). **(iv) SEC-DAPO:** SEC-style weighting combined with DAPO.

**Metrics:** Our primary metric is *worst-task accuracy*

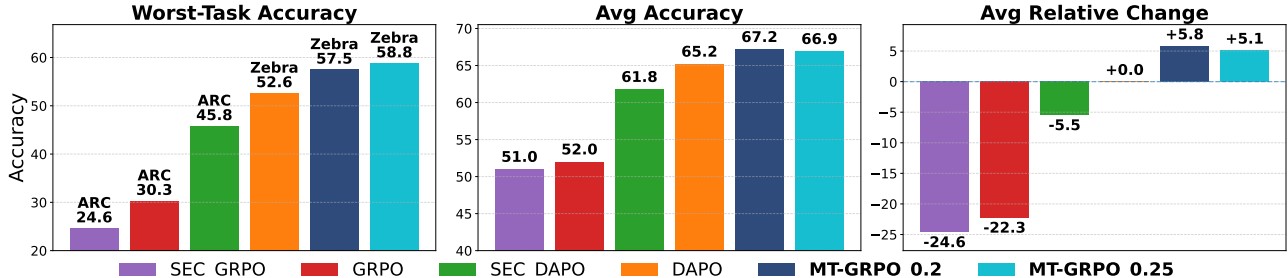

*Figure 4.* Experiment 1: MT-GRPO substantially outperforms all baselines in terms of worst-task accuracy by 6% or more without conceding on average accuracy. Moreover, it achieves higher average per-task relative change, reflecting stronger improvements on weaker tasks.

(minimum accuracy across tasks), which reflects robustness. To assess the robustness vs. overall performance trade-off, we also report *average accuracy* and *average per-task relative change* ((Liu et al., 2023; Navon et al., 2022), which normalizes gains across heterogeneous task scales: $\Delta m\% = \frac{1}{K} \sum_{k=1}^{K} \frac{\text{Acc}_m(k) - \text{Acc}_b(k)}{\text{Acc}_b(k)} \times 100$, where $b$ denotes the DAPO baseline. This normalization emphasizes gains on lower-performing tasks. Full training details are provided in Section E.

### 6.1. Experiment 1: Controlled Three-Task Setting

We evaluate MT-GRPO in a controlled multi-task setting by post-training Qwen-2.5-3B on three medium-difficulty tasks (Countdown, Zebra, ARC) and study how adaptive task reweighting affects training dynamics. An ablation study of the individual components of MT-GRPO is provided in Section D.2.

**Main results:** Figure 4 summarizes performance across methods. MT-GRPO achieves substantially higher worst-task accuracy than all baselines for both $\lambda = 0.2$ and $\lambda = 0.25$, while also improving average accuracy. These gains arise because MT-GRPO reallocates optimization effort away from the high-performing Countdown task toward weaker tasks. MT-GRPO also attains the highest average per-task relative change, indicating more balanced improvements across tasks. Consistent with the design in Subroutine 1, increasing $\lambda$ strengthens worst-task performance. $\lambda = 0.25$ yields higher worst-task accuracy than $\lambda = 0.2$, while $\lambda = 0.2$ achieves better average accuracy.

**Weight dynamics:** Figure 6 illustrates how MT-GRPO achieves these gains. Early in training, Zebra outperforms Countdown, but this reverses after approximately 50 steps. MT-GRPO responds by reallocating weight toward Zebra and reducing emphasis on Countdown, whereas DAPO and SEC-DAPO continue to prioritize Countdown even after it attains high performance. As a result, these baselines underperform on Zebra or ARC and achieve little additional improvement on Countdown, leading to poorer worst-task performance and lower average accuracy.

**Ratio Preservation:** Tasks with high prevalence of zero-gradient samples (e.g., ARC; Figure 3) tend to be substan-

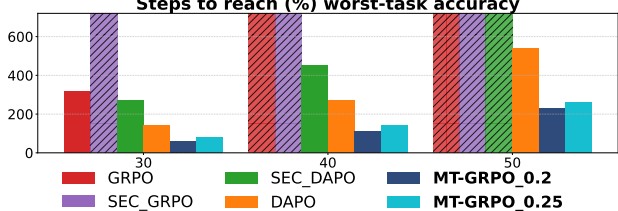

*Figure 5.* MT-GRPO reaches target worst-task accuracy thresholds substantially faster (50% fewer training steps) than baselines; striped bars indicate the threshold was not reached.

tially underrepresented in training batches relative to their intended weights (Figure 6). Our proposed RP SAMPLER (Algorithm 2) ensures that realized batch proportions closely track the learned task weights from Subroutine 1 and plays a critical role in the robust performance of Algorithm 1.

**Faster robustness gains:** Figure 5 reports the number of training steps required to reach specified worst-task accuracy thresholds. MT-GRPO consistently reaches these thresholds in fewer steps than all baselines. In several cases, baselines fail to reach the target thresholds within the training budget as marked in full length striped bars. This shows that beyond improving final worst-task accuracy, MT-GRPO also accelerates progress on the weakest tasks.

### 6.2. Experiment 2: Scaling to Nine Tasks

We evaluate whether the gains from Experiment 1 (Section 6.1) persist in a larger multi-task setting with increased task diversity by post-training Qwen-2.5-3B on nine tasks corresponding to the easy, medium, and hard variants of Countdown, Zebra, and ARC.

**Trade-off controlled by $\lambda$:** Figure 7 summarizes aggregate performance across methods for different values of the trade-off parameter $\lambda$. Increasing $\lambda$ consistently improves worst-task accuracy (by 16% over GRPO and 6% over DAPO at $\lambda = 1.2$) but reduces average accuracy, demonstrating how the trade-off encoded in our objective (Equation (5)) is operationalized by the task-reweighting mechanism in Algorithm 1 and subroutine 1. Specifically, larger $\lambda$ concentrates gains on the worst-performing task (Zebra-hard), while smaller $\lambda$ promotes more balanced progress across tasks, yielding higher average relative

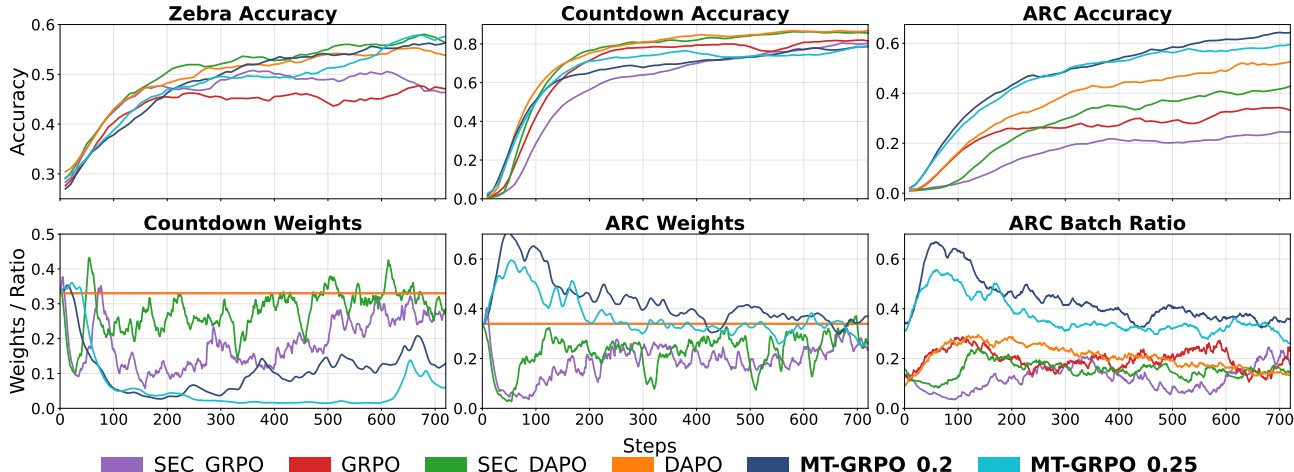

*Figure 6.* Experiment 1: **Top plots:** Task-wise accuracies. **Bottom plots:** MT-GRPO reallocates weights toward the under-performing tasks. In contrast, baselines continue to prioritize high-performing Countdown, leading to weaker performance on Zebra or ARC and only marginal gains on Countdown. ARC is typically underrepresented in training batches relative to its task weight (middle vs right). RP SAMPLER ensures alignment of realized batch proportions with intended task weights and facilitates higher ARC performance.

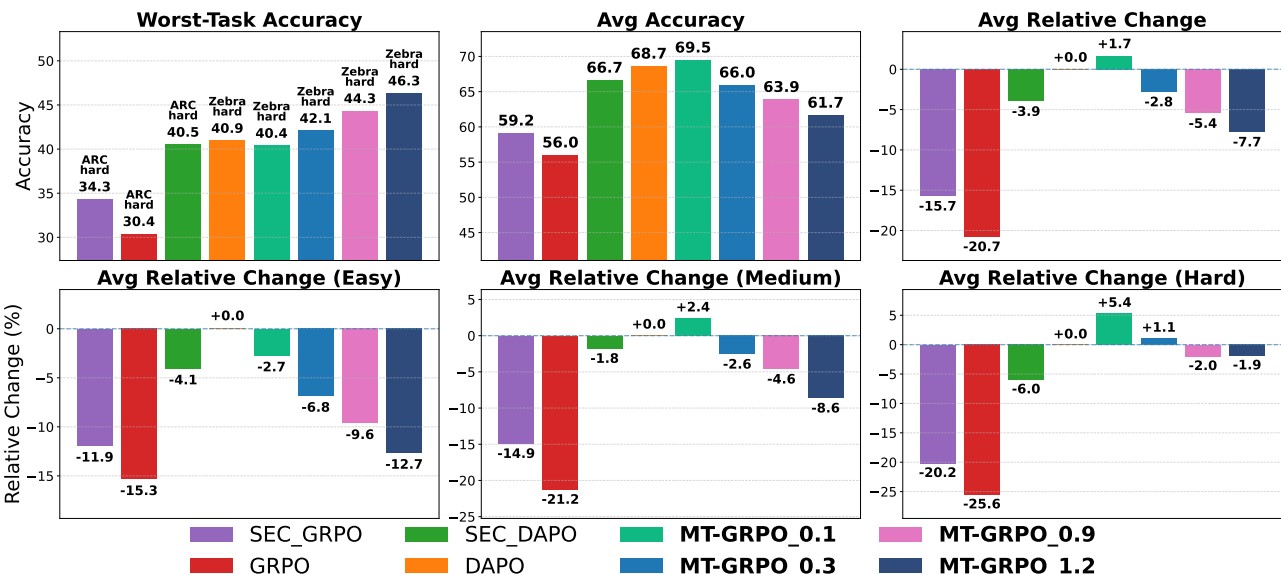

*Figure 7.* Experiment 2: **Top plots:** Increasing $\lambda$ improves worst-task accuracy of MT-GRPO but reduces average accuracy, highlighting a trade-off controlled by $\lambda$. $\lambda = 0.1$ yields the highest average per-task relative change, showing stronger gains on weaker tasks. **Bottom plots:** For smaller $\lambda$ (0.1, 0.3), MT-GRPO prioritizes slower-improving tasks, yielding larger gains on hard tasks while sacrificing easy ones. For larger $\lambda$, gains concentrate on the lowest-performing task, increasing worst-task accuracy but reducing average relative change.

change. These results confirm that $\lambda$ provides effective control over this trade-off in practice.

**Difficulty-wise analysis:** Figure 7 (bottom plots) reports average per-task relative change by difficulty. For smaller $\lambda$, MT-GRPO exhibits negative relative change on easy tasks but positive relative change on hard tasks, indicating a reallocation of optimization effort toward more challenging tasks. This is expected because smaller $\lambda$ places greater emphasis on improvement (Subroutine 1), and harder tasks typically exhibit slower learning progress. This also explains the higher overall average per-task relative change observed for smaller $\lambda$, as this metric is more sensitive to improvements

on weaker tasks than on high-performing ones.

**Comparison to sequential training:** We also compare against training each task family in sequence, a natural alternative to joint multi-task training. On the same nine-task setting, we train over Countdown, Zebra, and ARC for 240 steps per family for two different orderings, using both GRPO and DAPO. Notably, even though sequential training specializes on one task family at a time, our jointly trained MT-GRPO ($\lambda$=1.2) surpasses its strongest variant on worst-task accuracy (46.3% vs. 41.1%). Moreover, we observe that sequential training is also sensitive to the task ordering and prone to forgetting earlier task families as train-

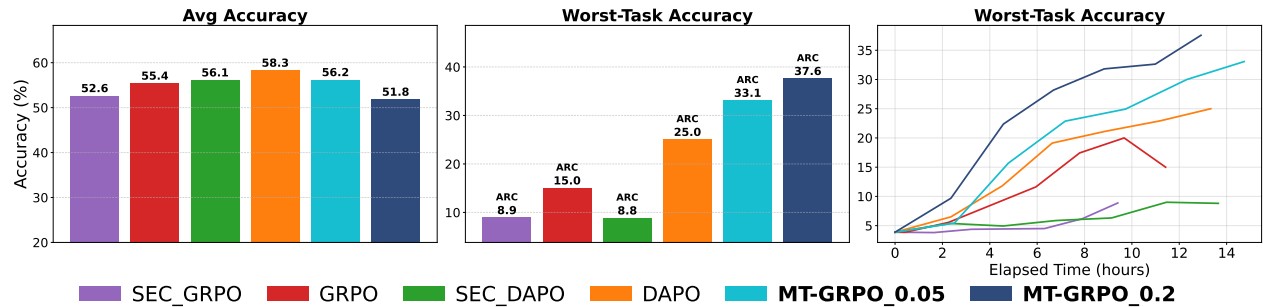

*Figure 8.* Experiment 3 (Qwen2.5-7B on MATH + ARC + SciKnowEval): MT-GRPO scales to larger models and realistic, domain-diverse task mixtures. MT-GRPO ($\lambda$=0.2) achieves a 12.6% worst-task accuracy gain over DAPO and reaches DAPO's final worst-task accuracy of 25.0% in ∼60% less training time (5.5 vs 13.3 hours).

ing progresses, making it harder to deploy in practice (see Figure 15). We defer the detailed analysis to Section D.3.

### 6.3. Experiment 3: Generalization to Larger Models and Diverse Tasks

To assess scalability to larger models and more realistic task families, we evaluate MT-GRPO with Qwen2.5-7B on a heterogeneous mixture of MATH, ARC, and SciKnowEval, and with OLMo-3 7B on SciKnowEval (Section D.1).

**Qwen2.5-7B on heterogeneous tasks:** We post-train Qwen2.5-7B (Qwen et al., 2024) on a diverse mixture of MATH (Hendrycks et al., 2021), ARC, and SciKnowEval (chemistry, physics) for 150 steps, spanning mathematical reasoning, inductive reasoning, and natural-language QA. Figure 8 shows that MT-GRPO substantially improves worst-task accuracy (ARC) and yields more balanced optimization across tasks, with a 12.6% gain over the strongest baseline (DAPO) when $\lambda$=0.2. Moreover, it reaches DAPO's final worst-task accuracy (25%) in ∼5.5 hours compared to DAPO's ∼13.3 hours, a ∼60% reduction in training time (Figure 8, right). As in Experiments 1–2, $\lambda$ acts as the trade-off knob: smaller $\lambda$ favours average accuracy while larger $\lambda$ prioritises worst-task robustness.

Across all three experiments, MT-GRPO consistently improves worst-task performance while maintaining competitive average accuracy. In the controlled setting (Experiment 1), weight-dynamics analysis shows these gains arise from adaptive reallocation of optimization effort to weaker tasks. Experiment 2 demonstrates scaling to nine heterogeneous tasks, and Experiment 3 confirms generalization to 7B models and diverse task mixtures. The trade-off parameter $\lambda$ provides effective control over robustness vs. average performance throughout. Together, these results validate both the empirical effectiveness of MT-GRPO and the intended behavior of its underlying optimization mechanism.

## 7. Conclusion

We introduced MULTI-TASK GRPO (MT-GRPO), a robustness-aware post-training algorithm for improving LLM reasoning across tasks. MT-GRPO performs improvement-aware task reweighting to promote balanced progress across tasks, and ratio-preserving sampling to ensure task weights translate into actual gradient contributions. Our experiments demonstrate that MT-GRPO consistently improves worst-task performance while maintaining competitive average accuracy. These results suggest that explicitly optimizing for task-wise robustness is practical and beneficial for building general-purpose reasoning models.

**Limitations:** MT-GRPO aims to improve robustness within the training task mixture, and extending it to be robust even w.r.t. tasks outside the mixture is an important direction for future work. Moreover, our training pipeline relies on verifiable rewards, under which the task reweighting mechanism is reliable. Characterizing MT-GRPO in non-verifiable domains remains an open direction.

## Acknowledgements

Ilija Bogunovic was supported by the EPSRC New Investigator Award EP/X03917X/1. Sangwoong Yoon was supported by Institute of Information & Communications Technology Planning & Evaluation (IITP) grant funded by the Korea government (MSIT) (No.RS-2020-II201336, Artificial Intelligence Graduate School Program (UNIST); No.RS-2025-25442824, AI Star Fellowship Program (UNIST)), the Center for Advanced Computation at Korea Institute for Advanced Study, and the InnoCORE program of the Ministry of Science and ICT (1.260017.01).

## Impact Statement

Given the increasing reliance on LLMs across various domains, including healthcare, education, legal systems, and autonomous decision-making, exhibiting broad competence across diverse reasoning skills is crucial for their deploy-

ment. This work contributes to the reliable deployment of large language models (LLMs) by developing MT-GRPO, an RL post-training algorithm that promotes balanced reasoning capabilities across tasks.

Moreover, we emphasize that our method is designed specifically to enhance AI reasoning capabilities, and as such, we do not anticipate any potential for unethical applications.

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

# A. Related Work

Recent post-training has produced strong results on individual reasoning benchmarks, but typical pipelines still optimize tasks largely in isolation, producing specialist models rather than balanced competence across heterogeneous reasoning skills. Naive multi-task post-training can further suffer from task interference and negative transfer, motivating robustness-aware and multi-objective perspectives for improving worst-task behavior without sacrificing overall performance. The related work below summarizes (i) robustness objectives and distributional robustness, (ii) multi-task optimization methods for mitigating gradient conflict, and (iii) multi-task LLM training and RL post-training (including GRPO) in both single- and multi-task settings.

**Robust objectives and distributional robustness.** Robust learning objectives that balance average performance with guarantees on underperforming groups or domains have a long history in distributionally robust optimization (DRO) (Namkoong & Duchi, 2016; Sagawa et al., 2019). Beyond worst-case group robustness, DRO is also closely connected to risk-sensitive objectives that trade off mean performance and variability (Gotoh et al., 2018), and recent work has developed non-asymptotic theory for DRO in modern non-convex regimes (Jin et al., 2021) as well as generalized formulations such as kernel DRO (Zhu et al., 2021). In large-scale language model training, robustness over mixed data has been pursued via domain reweighting and mixture optimization methods (Oren et al., 2019; Xie et al., 2023; Liu et al., 2024b; Grangier et al., 2024; Diao et al., 2025; Hu et al., 2024), as well as gradient-aware approaches for improved alignment across domains (Fan et al., 2023; 2025). These efforts primarily target pre-training or supervised objectives over heterogeneous corpora, and do not directly instantiate robustness-aware objectives within RL post-training for LLMs, where the optimization signal is mediated by task-dependent rewards and on-policy sampling.

**Multi-task optimization and gradient conflict methods.** A central difficulty in multi-task learning is that task gradients may conflict or exhibit large magnitude disparities, leading to updates that degrade some tasks despite improving others (Yu et al., 2020). This has motivated recent works on multi-objective and multi-task optimization, including classical multiple-gradient descent methods (Désidéri, 2012) and their deep learning instantiations that cast MTL as multi-objective optimization (Sener & Koltun, 2018), task balancing via gradient normalization (Chen et al., 2018), conflict-aware or constrained updates (Liu et al., 2021a), gradient manipulation methods (Chen et al., 2020; Kurin et al., 2022; Liu et al., 2022; Zhu et al., 2025a), and game-theoretic or bargaining-style formulations (Navon et al., 2022). A complementary line of work targets negative transfer via geometry-aware gradient balancing and homogenization (Liu et al., 2021b; Javaloy & Valera, 2021; Wang et al., 2020). Other approaches emphasize task weighting rules (e.g., uncertainty-based loss weighting) as a lightweight mechanism for balancing objectives (Kendall et al., 2018), while objectives that encourage progress on the worst-improving task provide another robustness-inspired alternative (Liu et al., 2023). Complementary results characterize when scalarization may fail to recover the full Pareto front (Hu et al., 2023), and analyze convergence issues of stochastic multi-objective methods along with stabilizing schemes (Zhou et al., 2022; Xiao et al., 2023). However, these approaches are typically studied in supervised multi-task settings and do not address RL post-training peculiarities such as task-dependent zero-gradient rates and the limited reliability of RL losses as direct proxies for task-level performance.

**Multi-task fine-tuning and RL post-training** Multi-task supervised fine-tuning is widely used to share statistical strength across related tasks, particularly when some tasks are data-limited; for example, (Liu et al., 2024a) studies multi-task fine-tuning for coding, and broader multi-task learning frameworks have been explored in related settings (Zhang et al., 2023; Eide & Frigessi, 2024; Gong et al., 2024; Wang et al., 2024; Feng et al., 2024; Qi et al., 2024; Brief et al., 2024; Wang et al., 2023; Zhu et al., 2025b). RL fine-tuning of LLMs has been extensively used for preference alignment (Christiano et al., 2017; Ziegler et al., 2019; Bai et al., 2022; Ouyang et al., 2022), for reasoning improvements with task rewards (Shao et al., 2024), and for self-training with internal supervision (Zelikman et al., 2022); more recently, GRPO-based post-training has shown strong reasoning performance (Lambert et al., 2024; Guo et al., 2025; Yue et al., 2025; Chen et al., 2025a; Zheng et al., 2025; Liu et al., 2025; Jin et al., 2025; Huang et al., 2025). A few works analyze and adapt GRPO to the multi-reward setting (Zhong et al., 2025; Liu et al., 2026; Lu et al., 2025). Only a few works study multi-task RL post-training directly, including mixture selection for higher average performance (Akter et al., 2025; Liang et al., 2025), cross dataset reward normalization to balance reward scales across datasets (Su et al., 2025), applying GRPO to a curated set of temporal tasks, text image translation skills (Wu et al., 2025b; Feng et al., 2025), sequential pipelines and task ordering to mitigate forgetting (Pang et al., 2025; Li et al., 2025), discussion about utility of meta-reasoning frameworks (Yan et al., 2025), and analyses of gradient imbalance and the limits of gradient-based curricula (Wu et al., 2025a). Curriculum sampling approaches primarily adjust task/difficulty sampling to improve efficiency or average outcomes (Wang et al., 2025; Zhang et al., 2025; Parashar

et al., 2025; Chen et al., 2025b), while transfer analyses evaluate whether gains generalize beyond the training domain (Huan et al., 2025). In contrast, our work incorporates task-wise robustness directly into the multi-task RL objective under GRPO, and introduces mechanisms such as adaptive reweighting and ratio-preserving batch construction to improve worst-task performance while maintaining strong overall performance.

# B. Background

This section provides additional background on KL-regularized policy optimization and the practical GRPO objective used in our experiments.

**KL-regularized objective.** Given a dataset of prompts $x \sim D$, a reward function $R(x, y)$, and a reference policy $\pi_{\mathrm{ref}}$, the KL-regularized objective for a policy $\pi_\theta$ is

$$J(\theta) = \mathbb{E}_{x \sim D}\Big[\mathbb{E}_{y \sim \pi_\theta(\cdot \mid x)}[R(x, y)] - \beta\, D_{\mathrm{KL}}(\pi_\theta(\cdot \mid x) \,\|\, \pi_{\mathrm{ref}}(\cdot \mid x))\Big], \tag{9}$$

where $\beta > 0$ controls the strength of regularization.

**Policy gradient.** The gradient of the KL-regularized objective can be written using the policy gradient theorem as

$$\nabla_\theta J(\theta) = \mathbb{E}_{x, y \sim \pi_\theta}\Big[\nabla_\theta \log \pi_\theta(y \mid x)\big(R(x, y) - \beta \log \tfrac{\pi_\theta(y|x)}{\pi_{\mathrm{ref}}(y|x)}\big)\Big]. \tag{10}$$

In practice, this gradient is estimated using Monte Carlo samples from the policy, and its variance is reduced by replacing rewards with advantage estimates.

**GRPO objective (prompt-level form).** Group Relative Policy Optimization (GRPO) samples, for each prompt $x$, a group of $G$ responses $\{y_i\}_{i=1}^G$ from the old policy $\pi_{\theta_{\mathrm{old}}}$. A prompt-level relative advantage is computed via within-group normalization:

$$A(x, y_i) = A(x, y_i) = \frac{\Big(R(x, y_i) - \mathrm{mean}(\{R(x, y_j)\}_{j=1}^G)\Big)}{\mathrm{std}(\{R(x, y_j)\}_{j=1}^G)} \tag{11}$$

Using importance sampling, the GRPO objective to update the current policy $\pi_\theta$ is

$$J_{\mathrm{GRPO}}(\theta) = \mathbb{E}_x\left[\mathbb{E}_{\{y_i\} \sim \pi_{\theta_{\mathrm{old}}}}\Big[\frac{1}{G}\sum_{i=1}^G \rho_i\Big(A(x, y_i) - \beta \log \tfrac{\pi_\theta(y_i|x)}{\pi_{\mathrm{ref}}(y_i|x)}\Big)\Big]\right], \tag{12}$$

where $\rho_i = \pi_\theta(y_i \mid x)/\pi_{\theta_{\mathrm{old}}}(y_i \mid x)$ is the importance ratio.

**Token-level formulation with clipping.** In practice, optimization is performed at the token level. Let $y_{i,t}$ denote the $t$-th token of response $y_i$, and define

$$r_{i,t} = \frac{\pi_\theta(y_{i,t} \mid x, y_{i,<t})}{\pi_{\theta_{\mathrm{old}}}(y_{i,t} \mid x, y_{i,<t})}.$$

We set the token-level advantages $\hat{A}_{i,t}$, as the same end of output normalized rewards,

$$A(x, y_i) = \frac{\Big(R(x, y_i) - \mathrm{mean}(\{R(x, y_j)\}_{j=1}^G)\Big)}{\mathrm{std}(\{R(x, y_j)\}_{j=1}^G)}.$$

In order to avoid instability, the importance ratios are usually clipped obtaining the below clipped GRPO objective:

$$J_{\mathrm{GRPO}}(\theta) = \mathbb{E}_{x, \{y_i\}}\left[\frac{1}{G}\sum_{i=1}^G \frac{1}{|y_i|}\sum_{t=1}^{|y_i|}\Big(\min\big(r_{i,t}\hat{A}_{i,t},\, \mathrm{clip}(r_{i,t}, 1-\varepsilon, 1+\varepsilon)\hat{A}_{i,t}\big) - \beta\, r_{i,t}\, f_{\mathrm{KL}}\big(\tfrac{\pi_{\mathrm{ref}}(y_{i,t}|x, y_{i,<t})}{\pi_\theta(y_{i,t}|x, y_{i,<t})}\big)\Big)\right], \tag{13}$$

where $\varepsilon$ is the PPO-style clipping threshold and

$$f_{\mathrm{KL}}(u) = u - \log u - 1. \tag{14}$$

# C. Proofs and Theoretical Insights

In this section, we detail all the main derivations and proofs used in Section 2.

## C.1. Closed form of induced regularizer $\Omega(z)$

In this subsection, we derive the closed form expression for the induced regularizer $\Omega(z)$ starting from the original constrained optimization problem in Equation (4).

For the purposes of analysis, we note that the absolute-value constraint is equivalent to the pair of linear constraints $J_k(\theta) - J_j(\theta) \leq \varepsilon$ and $J_j(\theta) - J_k(\theta) \leq \varepsilon$. Hence, we start from the simplified constrained problem

$$\max_{\theta \in \Theta} \quad \frac{1}{K} \sum_{k=1}^{K} J_k(\theta) \quad \text{s.t.} \quad J_k(\theta) - J_j(\theta) \leq \varepsilon, \ \forall (k,j) \in [K] \times [K]. \tag{15}$$

Here, the constraints with $k = j$ are trivial, but we keep the full indexing for notational convenience.

**Lagrangian Reformulation:** We introduce dual variables $\mu_{kj} \geq 0$ for each constraint $J_k(\theta) - J_j(\theta) \leq \varepsilon$. giving the Lagrangian

$$\mathcal{L}(\theta, \mu) = \frac{1}{K} \sum_{k=1}^{K} J_k(\theta) - \sum_{k=1}^{K} \sum_{j=1}^{K} \mu_{kj} \Big( J_k(\theta) - J_j(\theta) - \varepsilon \Big). \tag{16}$$

Collecting terms with respect to $J_k(\theta)$ yields

$$\mathcal{L}(\theta, \mu) = \sum_{k=1}^{K} \Big( \frac{1}{K} - \sum_{j=1}^{K} \mu_{kj} + \sum_{j=1}^{K} \mu_{jk} \Big) J_k(\theta) + \varepsilon \sum_{k=1}^{K} \sum_{j=1}^{K} \mu_{kj}. \tag{17}$$

Note, each $J_k(\theta)$ is now weighted by a factor,

$$\tilde{z}_k(\mu) := \frac{1}{K} - \sum_{j=1}^{K} \mu_{kj} + \sum_{j=1}^{K} \mu_{jk}, \qquad k \in [K]. \tag{18}$$

Rewriting Equation (17) with $z_k(\mu)$, we have,

$$\mathcal{L}(\theta, \mu) = \sum_{k=1}^{K} \tilde{z}_k(\mu) J_k(\theta) + \varepsilon \sum_{k=1}^{K} \sum_{j=1}^{K} \mu_{kj}. \tag{19}$$

Here, note that $\sum_{k=1}^{K} \tilde{z}_k(\mu) = 1$ for all $\mu$:

$$\sum_{k=1}^{K} \tilde{z}_k(\mu) = \sum_{k=1}^{K} \frac{1}{K} - \sum_{k=1}^{K} \sum_{j=1}^{K} \mu_{kj} + \sum_{k=1}^{K} \sum_{j=1}^{K} \mu_{jk} = 1. \tag{20}$$

This yields the saddle form,

$$\max_{\theta \in \Theta} \min_{z \in \Delta_K} \sum_{k=1}^{K} z_k J_k(\theta) + \varepsilon \, \Omega(z), \tag{21}$$

where $\Omega$ is the regularizer induced by eliminating $\mu$.

**Eliminating $\mu$ and defining $\Omega(z)$.** The mapping $\mu \mapsto z$ is not injective as multiple $\mu$ values can induce the same $z$. Hence, $\sum_{k,j} \mu_{kj}$ is not uniquely determined by $z$. We therefore define the induced regularizer as the *minimum* total dual mass among all $\mu$ that induce the given $z$:

$$\Omega(z) := \min_{\mu \geq 0} \Big\{ \sum_{k=1}^{K} \sum_{j=1}^{K} \mu_{kj} \ : \ z_k = \frac{1}{K} - \sum_{j=1}^{K} \mu_{kj} + \sum_{j=1}^{K} \mu_{jk}, \ \forall k \Big\}. \tag{22}$$

Let $r_k := \sum_{j=1}^K \mu_{kj}, c_k := \sum_{j=1}^K \mu_{jk}$ and substituting it in the constraints of Equation (22), we have

$$r_k - c_k = \frac{1}{K} - z_k =: d_k, \qquad k \in [K]. \tag{23}$$

Moreover, note that $\sum_{k=1}^K d_k = 1 - \sum_k z_k = 0$, using which the objective in Equation (22) can be written as

$$\sum_{k=1}^K \sum_{j=1}^K \mu_{kj} = \sum_{k=1}^K r_k. \tag{24}$$

Thus, Equation (22) is equivalent to the minimum-flow problem

$$\Omega(z) = \min_{\mu \geq 0} \sum_{k=1}^K r_k \quad \text{s.t.} \quad r_k - c_k = d_k, \ \forall k. \tag{25}$$

Interpreting $\mu_{kj}$ as flow shipped from node $k$ to node $j$ on a complete directed graph, $d_k > 0$ are supplies and $d_k < 0$ are demands. Any feasible flow requires that the total flow is equal to or greater than the total supply:

$$\sum_{k=1}^K \sum_{j=1}^K \mu_{kj} = \sum_{k=1}^K r_k \geq \sum_{k:\, d_k > 0} (r_k - c_k) = \sum_{k:\, d_k > 0} d_k. \tag{26}$$

Conversely, since the graph is complete, the lower bound Equation (26) is tight as given the supplies required for each node $d_k$, one can always adapt $\mu$ to change flow from supply nodes to demand nodes such that the total shipped flow equals the total supply (e.g., via a greedy matching construction). Hence the lower bound Equation (26) is tight and

$$\Omega(z) = \sum_{k:\, d_k > 0} d_k. \tag{27}$$

Using $\sum_k d_k = 0$, we have $\sum_{d_k > 0} d_k = \frac{1}{2} \sum_{k=1}^K |d_k|$, so

$$\Omega(z) = \frac{1}{2} \sum_{k=1}^K |d_k| = \frac{1}{2} \sum_{k=1}^K \left| z_k - \frac{1}{K} \right| = \frac{1}{2} \left\| z - \frac{1}{K} \mathbf{1} \right\|_1. \tag{28}$$

Equation (28) shows that the induced regularizer penalizes deviations from uniform task weights via an $\ell_1$ distance.

### C.2. Update rule for strict worst task reward maximization

Here, we detail the update rule derived for strict worst-task reward maximization in Equation (6).

Specifically, we parameterize the task weights $z \in \Delta_K$ using a softmax over logits $\xi \in \mathbb{R}^K$, $z_t = \mathrm{Softmax}(\xi_t)$, and perform gradient descent on the inner objective in Equation (5) (with $\varepsilon = 0$) for fixed $\theta$ w.r.t. $\xi_t$. At iteration $t$, the inner objective is $L(\xi_t) = \sum_{k=1}^K z_{k,t} J_k(\theta_t)$. Taking gradients with respect to $\xi_t$, we obtain

$$g_t = \nabla_{\xi_t} L(\xi_t) = \left( \frac{\partial z_t}{\partial \xi_t} \right)^\top J(\theta_t)$$

$$\implies (g_t)_k = z_{k,t} \left( J_k(\theta_t) - \sum_{j=1}^K z_{j,t} J_j(\theta_t) \right). \tag{29}$$

where $J(\theta_t) = [J_1(\theta_t), \ldots, J_K(\theta_t)]^\top$. The logits $\xi_t$ are updated using $g_t$, which increases the weight on tasks whose reward is below the current weighted average and allows unconstrained optimization over $\xi$ while ensuring $z_t \in \Delta_K$.

Subsequently at iteration $t$, the resulting updates are given by

$$\theta_{t+1} = \theta_t + \gamma_t \sum_{k=1}^K z_k^t \nabla_\theta J_{\mathrm{GRPO},k}(\theta_t), \qquad \xi_{t+1} = \xi_t - \beta \, g_t, \quad \text{where} \quad g_t = \begin{bmatrix} \nabla_\xi z_{1,t}^\top \\ \vdots \\ \nabla_\xi z_{K,t}^\top \end{bmatrix} \begin{bmatrix} J_1(\theta_t) \\ \vdots \\ J_K(\theta_t) \end{bmatrix}, \tag{30}$$

Next, we derive the exact gradient expression in Equation (29) in detail.

Recall that task weights are parameterized via logits $\xi \in \mathbb{R}^K$ using the softmax function: $z_k(\xi) = \frac{e^{\xi_k}}{\sum_{m=1}^K e^{\xi_m}}$. For fixed policy parameters $\theta_t$, we define the inner objective $L(\xi) := \sum_{k=1}^K z_k(\xi) J_k(\theta_t)$, and denote $J_k := J_k(\theta_t)$ for brevity. Let $\bar{J} := \sum_{j=1}^K z_j(\xi) J_j$ be the current weighted average reward.

**Softmax derivative.** The Jacobian of the softmax function satisfies the standard identity

$$\frac{\partial z_i}{\partial \xi_k} = z_i(\delta_{ik} - z_k),$$

where $\delta_{ik}$ is the Kronecker delta with $\delta_{ik} = 1$ if $i = k$ and $\delta_{ik} = 0$ when $i \neq k$.

**Gradient of the inner objective.** We compute the gradient of $L(\xi)$ with respect to $\xi_k$:

$$\frac{\partial L}{\partial \xi_k} = \sum_{i=1}^K \frac{\partial z_i}{\partial \xi_k} J_i = \sum_{i=1}^K z_i(\delta_{ik} - z_k) J_i$$

$$= z_k J_k - z_k \sum_{i=1}^K z_i J_i$$

$$= z_k(J_k - \bar{J}).$$

Thus, the gradient admits the closed form

$$(\nabla_\xi L(\xi))_k = z_k\left(J_k - \sum_{j=1}^K z_j J_j\right).$$

### C.3. Derivation of Improvement-Aware Task Reweighting

In this section, we concretely derive the improvement-aware update rule in Subroutine 1. Specifically, we define a per-step minimax optimization problem in terms of policy update direction $\theta_{t+1} - \theta_t$ and weights $z$ and use first-order Taylor approximation to jointly derive the policy update direction with task weights.

We begin by considering a generic policy update of the form

$$\theta_{t+1} = \theta_t + \gamma_t d_t, \tag{31}$$

where $d_t \in \mathbb{R}^m$ is the update direction that is dependent on $\nabla_\theta J_{\text{GRPO},k}(\theta_t)$ of all tasks $k$ and $\gamma_t > 0$ is a stepsize.

By first-order Taylor expansion,

$$I_k^{(t)} = J_{\text{GRPO},k}(\theta_t + \gamma_t d_t) - J_{\text{GRPO},k}(\theta_t)$$
$$= \gamma_t \langle \nabla_\theta J_{\text{GRPO},k}(\theta_t), d_t \rangle + \mathcal{O}(\gamma_t^2 \|d_t\|^2). \tag{32}$$

Thus, for sufficiently small $\gamma_t$, the improvement in task $k$ is proportional to the inner product between the task gradient $k$ and the update direction.

Our goal, then, is to define an optimal policy update direction $d_t$ and design an improvement-aware update strategy for $z$ such that we prioritize tasks that are underperforming or under-improving. To this end, we consider the following minimax objective at iteration $t$:

$$\max_{d_t \in \mathbb{R}^m} \min_{z \in \Delta_K} \frac{1}{\gamma_t} \sum_{k=1}^K z_k I_k^{(t)} - \frac{1}{2}\|d_t\|_2^2 + \lambda \sum_{k=1}^K z_k J_k(\theta_t), \tag{33}$$

and, using the first-order approximation in Equation (32), we obtain

$$\max_{d_t \in \mathbb{R}^m} \min_{z \in \Delta_K} \sum_{k=1}^K z_k \langle \nabla_\theta J_{\text{GRPO},k}(\theta_t), d_t \rangle - \frac{1}{2}\|d_t\|_2^2 + \lambda \sum_{k=1}^K z_k J_k(\theta_t), \tag{34}$$

where the quadratic penalty on $\|d_t\|_2^2$ keeps the update magnitude controlled, ensuring the validity of the first-order Taylor expansion.

Equation (34) balances worst-case improvement against worst-task performance while regularizing the policy update magnitude $d_t$ to reduce the first-order Taylor approximation error from Equation (32). The maximization over $d_t$ selects a policy update direction that best aligns with the tasks emphasized by $z$, while the inner minimization identifies tasks that are both underperforming or under-improving.

Note that the objective in Equation (34) is concave quadratic in $d_t$ and convex in $z$ with $z$ belonging to a compact set $\Delta_K$. Hence, strong duality holds (see Liu et al. (2023, Proposition 3.1)) and Equation (34) is equivalent to min-max swapped,

$$\min_{z \in \Delta_K} \max_{d_t \in \mathbb{R}^m} \sum_{k=1}^{K} z_k \langle \nabla_\theta J_{\mathrm{GRPO},k}(\theta_t), d_t \rangle - \frac{1}{2} \|d_t\|_2^2 + \lambda \sum_{k=1}^{K} z_k . J_k(\theta_t)$$

We can now solve the inner maximization by taking the gradient with respect to $d_t$ and setting it to zero, which yields the optimal $d_t$ as,

$$d_t^*(z) = \sum_{k=1}^{K} z_k \nabla_\theta J_{\mathrm{GRPO},k}(\theta_t). \tag{35}$$

Note that the term $\lambda J_k(\theta_t)$ vanishes when differentiating with respect to $d_t$, since it does not depend on $d_t$. Substituting Equation (35) back into Equation (34) eliminates $d_t$ and gives an equivalent minimization problem over $z$:

$$\min_{z \in \Delta_K} \frac{1}{2} \Big\| \sum_{k=1}^{K} z_k \nabla_\theta J_{\mathrm{GRPO},k}(\theta_t) \Big\|_2^2 + \lambda \sum_{k=1}^{K} z_k J_k(\theta_t). \tag{36}$$

Taking the gradient of the objective in Equation (36) with respect to $z$, we obtain for each $k \in [K]$,

$$\frac{\partial}{\partial z_k} \left[ \frac{1}{2} \Big\| \sum_{j=1}^{K} z_j \nabla_\theta J_{\mathrm{GRPO},j}(\theta_t) \Big\|_2^2 + \lambda \sum_{j=1}^{K} z_j J_j(\theta_t) \right] = \Big\langle \sum_{j=1}^{K} z_j \nabla_\theta J_{\mathrm{GRPO},j}(\theta_t), \nabla_\theta J_{\mathrm{GRPO},k}(\theta_t) \Big\rangle + \lambda J_k(\theta_t) \tag{37}$$

$$= \langle d_t, \nabla_\theta J_{\mathrm{GRPO},k}(\theta_t) \rangle + \lambda J_k(\theta_t) \tag{38}$$

$$\approx \gamma_t I_k^{(t)} + \lambda J_k(\theta_t), \qquad \text{using Equation (32).} \tag{39}$$

Equation (39) allows us to avoid computing inner products between per-objective gradients, which can be costly or noisy in practice. Instead, we use the per-step improvement approximation $I_k^{(t)}$ from Equation (32), yielding the surrogate signal

$$s_k^{(t)} := I_k^{(t)} + \lambda J_k(\theta_t), \tag{40}$$

where we absorb the factor $\gamma_t$ into the tunable parameter $\lambda$, as the learning rate $\gamma_t$ is constant in our training pipeline. Rather than fully solving Equation (36) at each step, we perform a single step gradient descent on $z$ using the surrogate signal in Equation (40).

As in Equation (6), to enforce $z \in \Delta_K$ during optimization, we parameterize $z_t = \mathrm{Softmax}(\xi_t), \xi_t \in \mathbb{R}^K$, and perform gradient descent with respect to $\xi_t$, which yields the update

$$\xi_{t+1} = \xi_t - \beta \begin{bmatrix} \nabla_\xi z_{1,t}^\top \\ \vdots \\ \nabla_\xi z_{K,t}^\top \end{bmatrix} \begin{bmatrix} I_1^{(t)} + \lambda J_1(\theta_t) \\ \vdots \\ I_K^{(t)} + \lambda J_K(\theta_t) \end{bmatrix}. \tag{41}$$

Using the softmax Jacobian identity $\frac{\partial z_i}{\partial \xi_k} = (\delta_{ik} - z_k)$, one obtains the closed form

$$\Big( \nabla_{\xi_t} \sum_{k=1}^{K} z_{k,t} s_k^{(t)} \Big)_k = z_{k,t} \Big( s_k^{(t)} - \sum_{j=1}^{K} z_{j,t} s_j^{(t)} \Big). \tag{42}$$

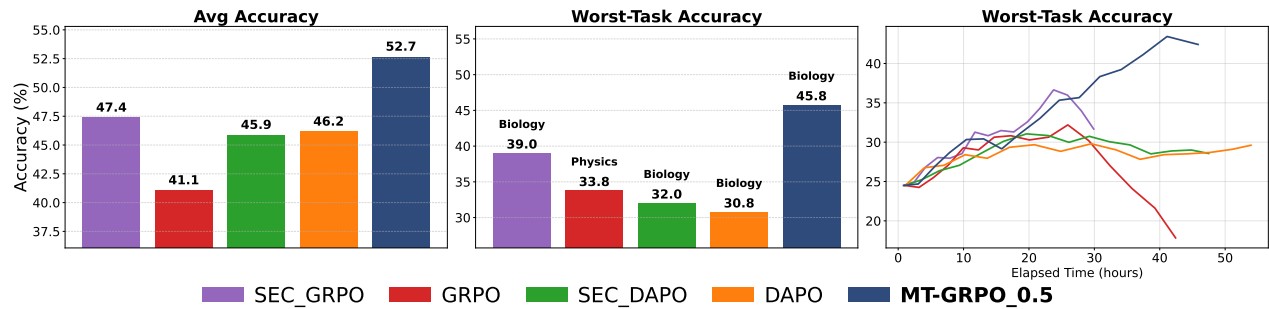

*Figure 9.* OLMo-3 7B on SciKnowEval. **Left/Middle:** average and worst-task accuracy at each method's best-worst-task step. **Right:** per-method worst-task training curves: baselines peak and then degrade, while MT-GRPO ($\lambda$=0.5) does not exhibit this degradation.

# D. Practical Implementation and Additional Experiments

This section provides supplementary experimental results and additional implementation analysis. Section D.1 presents the full details of Experiment 3 (Section 6.3), covering OLMo-3 7B on SciKnowEval and Qwen2.5-7B on a heterogeneous mixture of MATH, ARC, and SciKnowEval. Section D.2 provides an ablation isolating the contributions of IWU and RPS, evaluated on Experiments 1 and 2. Section D.4 contains additional analysis of RP SAMPLER , examining ratio-preserving sampling and acceptance-aware sampling.

## D.1. Experiment 3: Extended Evaluation on 7B-Scale Models

**OLMo-3 7B on SciKnowEval:** We post-train OLMo-3-7B-Instruct (Olmo et al., 2025) on SciKnowEval, a benchmark of natural-language QA tasks across four scientific domains: biology, chemistry, physics, and materials science. We train all methods for 150 steps and compare against GRPO, SEC-GRPO, DAPO, and SEC-DAPO. For MT-GRPO, we use $\lambda = 0.5$.

MT-GRPO ($\lambda$=0.5) achieves 45.8% worst-task accuracy, outperforming all baselines (Figure 9, middle). We report results at each method's best-worst-task step rather than the final training step, as several baselines peak and then degrade before training ends. In contrast, MT-GRPO does not exhibit this degradation, as shown in the right panel of Figure 9. This qualitative difference indicates more balanced optimisation across domains throughout training.

**Qwen2.5-7B on MATH + ARC + SciKnowEval:** We post-train Qwen2.5-7B (Qwen et al., 2024) on a heterogeneous mixture of MATH (Hendrycks et al., 2021), ARC, and SciKnowEval (chemistry and physics subtasks), spanning mathematical reasoning, inductive reasoning, and natural-language QA. We train all methods for 150 steps and compare against GRPO, SEC-GRPO, DAPO, and SEC-DAPO. For MT-GRPO, we report two values of the trade-off parameter, $\lambda \in \{0.05, 0.2\}$.

MT-GRPO substantially improves worst-task accuracy: 37.6% at $\lambda$=0.2 and 33.1% at $\lambda$=0.05, versus 25.0% for DAPO and 15.0% for GRPO. This is consistent with the $\lambda$-controlled trade-off seen in Experiment 2: larger $\lambda$ improves worst-task accuracy at the cost of average accuracy. Figure 8 visualises these results.

Together, these results demonstrate that MT-GRPO generalises to larger 7B-scale models and more realistic, domain-diverse task mixtures.

## D.2. Ablation Study

We isolate the two components of MT-GRPO: **GRPO_IWU** applies task reweighting only, without the ratio-preserving sampler, and **GRPO_RPS** applies the ratio-preserving sampler with fixed uniform task weights ($z = 1/K$), without adaptive reweighting. Results are reported on Experiment 1 (Figure 10) and Experiment 2 (Figure 11).

GRPO_IWU_0.25 improves worst-task accuracy from 30.3% to 41.9% via adaptive reweighting, while average accuracy drops from 52.0% to 43.9%. A similar pattern is visible in the 9-task setting (Figure 11). Moreover, GRPO_IWU_0.25 suffers from a mismatch between assigned task weights and effective gradient contributions, especially for ARC, the worst-performing task for GRPO_IWU_0.25, as it has a high zero-gradient filtering rate. Figure 12 confirms this as GRPO_IWU_0.25 assigns ARC a mean weight of ~0.80, while the realized post-filtered batch ratio is only ~0.45.

GRPO_RPS eliminates this mismatch by enforcing target task proportions in the post-filtered batch (Figure 13). In Experiment 1, it outperforms DAPO on both worst-task (54.6% vs 52.6%) and average accuracy (68.9% vs 65.2%).

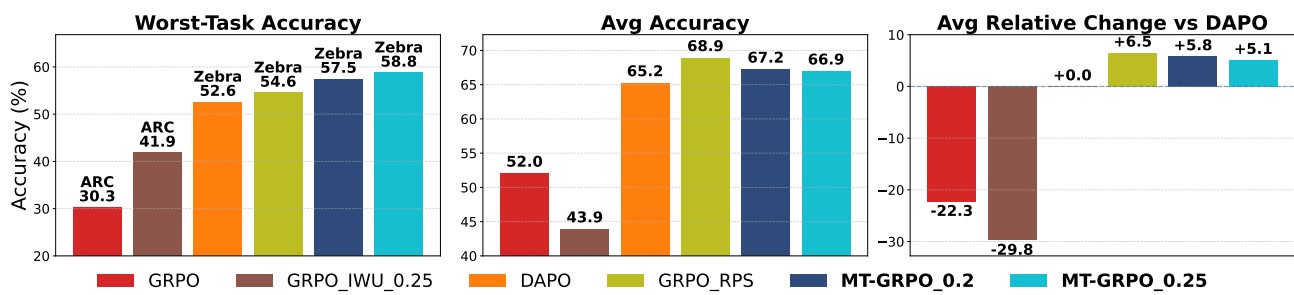

*Figure 10.* Ablation on Experiment 1 (3 tasks). GRPO_IWU_0.25 improves worst-task accuracy w.r.t. GRPO via adaptive reweighting but reduces average accuracy significantly. GRPO_RPS outperforms DAPO on both metrics. MT-GRPO (IWU + RPS) achieves the best worst-task accuracy with competitive average accuracy, showing the necessity of both components for balanced progress across tasks.

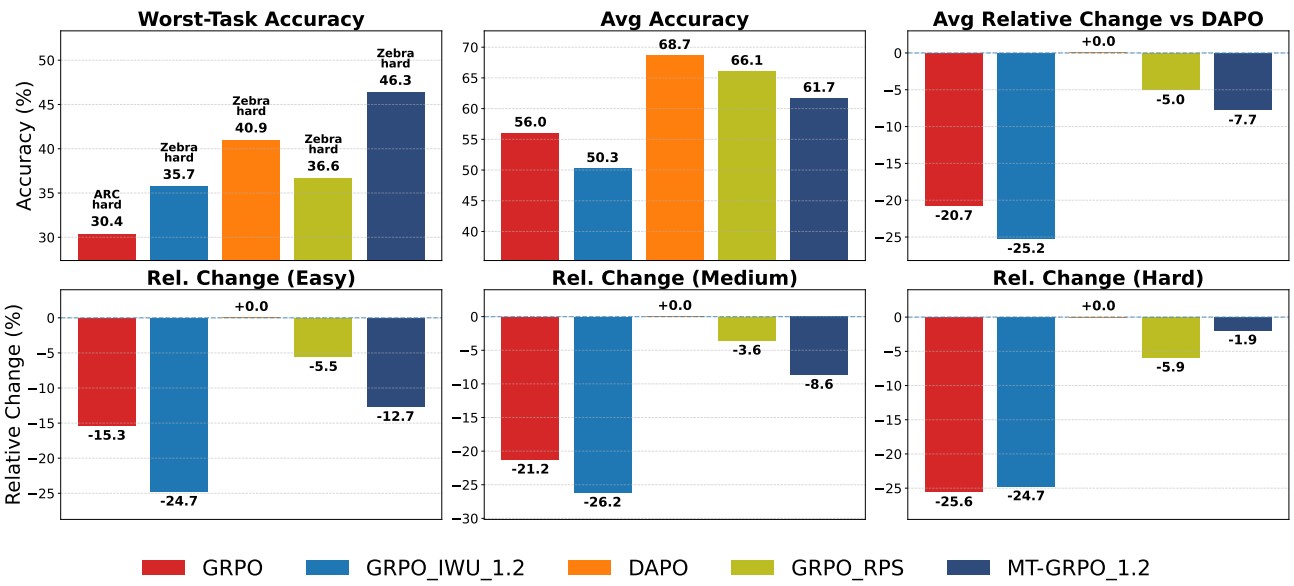

*Figure 11.* Ablation on Experiment 2 (9 tasks). **Top:** worst-task accuracy, average accuracy, and average relative change vs DAPO across ablation variants. **Bottom:** per-difficulty relative change. MT-GRPO ($\lambda$=1.2) achieves the best worst-task accuracy, reinforcing the necessity of both components for robust performance across tasks.

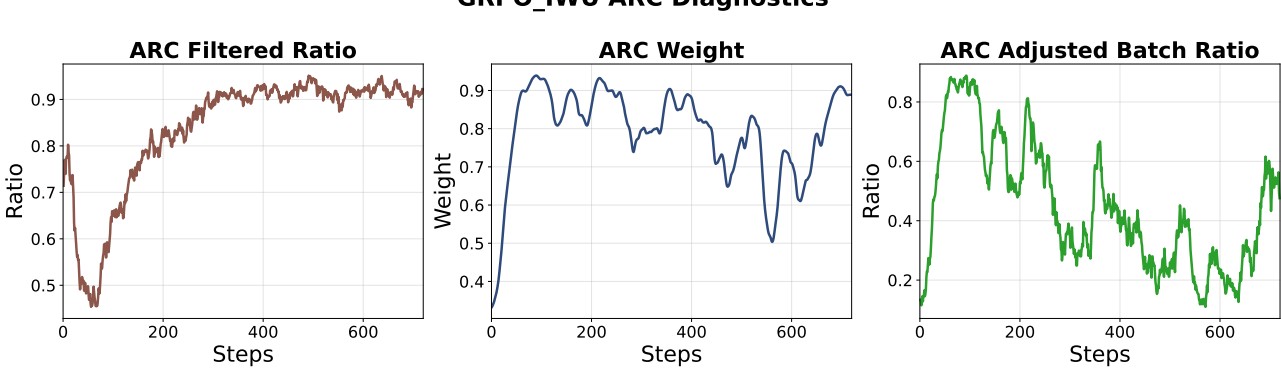

*Figure 12.* GRPO_IWU diagnostics for ARC (Experiment 1). **Left:** ARC filtered ratio (fraction of ARC prompts with zero advantage). **Middle:** task weight that IWU assigns to ARC. **Right:** realized ARC batch ratio after filtering. The gap between the assigned weight and the realized batch ratio illustrates the mismatch that RPS is designed to eliminate.

MT-GRPO combines both components, achieving the best worst-task accuracy in both experiments (58.8% at $\lambda$=0.25 in Experiment 1, 46.3% at $\lambda$=1.2 in Experiment 2) with competitive average accuracy.

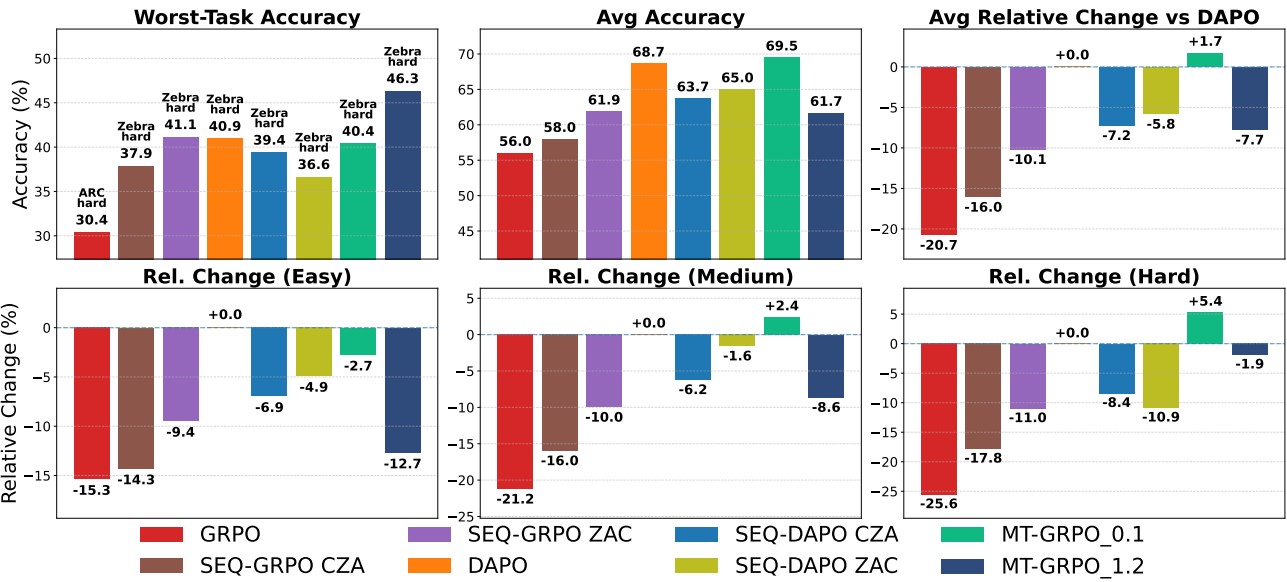

*Figure 13.* GRPO_RPS batch ratios (Experiment 1). With uniform task weights, the RP sampler maintains each of the three tasks at ∼1/3 of the post-filtered batch across training, despite their differing zero-gradient rates.

*Figure 14.* Sequential versus joint training on the nine-task Experiment 2 setting, training each family (Countdown, Zebra, ARC) for 240 steps in turn under two orderings (CZA, ZAC), with GRPO and DAPO (SEQ-GRPO, SEQ-DAPO). Sequential training underperforms MT-GRPO on worst-task accuracy across both GRPO and DAPO variants.

### D.3. Sequential Training: Per-Task Comparison and Forgetting

This section provides the per-task comparison and forgetting dynamics for the sequential-training baseline discussed in Section 6.2. We adopt a family-level sequential setup that merges all difficulty levels within a task and trains over Countdown, Zebra, and ARC for 240 steps each (720 steps total), under two orderings: Countdown→Zebra→ARC (CZA) and Zebra→ARC→Countdown (ZAC), using both GRPO and DAPO (SEQ-GRPO, SEQ-DAPO). Figure 14 compares all methods across the nine tasks: sequential training consistently underperforms MT-GRPO on worst-task accuracy, and the CZA and ZAC variants yield noticeably different outcomes. Figure 15 shows the forgetting dynamics, where accuracy on an earlier task family degrades once training switches to a later one, with the extent depending on the ordering. Together, this sensitivity to ordering and forgetting make sequential training non-trivial to deploy, since one must search over orderings to find the optimal one, whereas MT-GRPO optimizes all tasks jointly and is order-agnostic.

### D.4. Analysis of RP SAMPLER : Effect of RPS and Acceptance-Aware Sampling

In this section, we provide the full version of RP SAMPLER introduced in Section 5, along with additional experimental results from Experiments 1 and 2 that were deferred to the appendix due to space constraints.

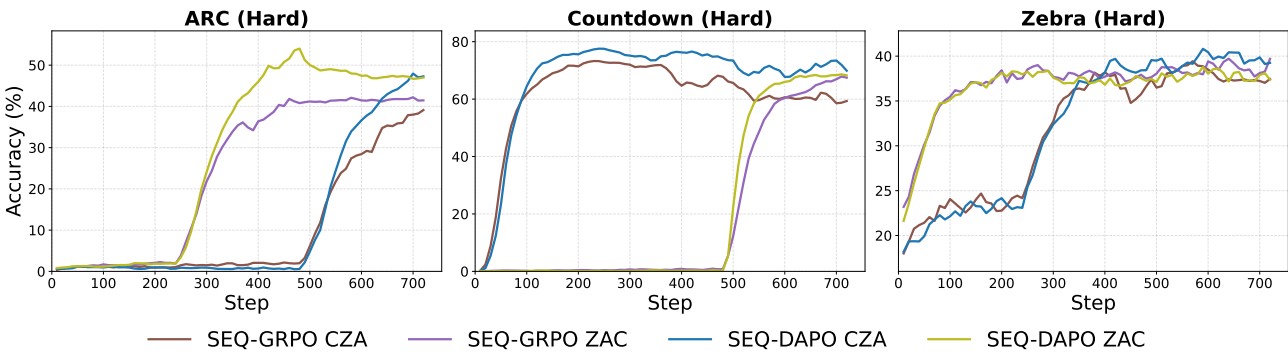

*Figure 15.* Signs of forgetting under sequential training. As training switches to a later family, accuracy on earlier families degrades, with the extent depending on the ordering. MT-GRPO avoids this by training all families jointly.

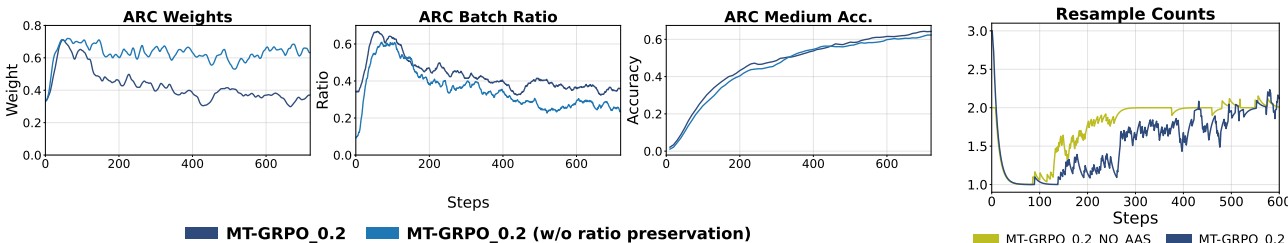

*Figure 16.* Effect of RP SAMPLER on ARC (Experiment 1). **First three panels:** MT-GRPO vs. a variant without the ratio-preserving constraint. Despite higher assigned weights, ARC's realized batch proportion is lower, yielding lower accuracy. **Rightmost panel:** removing Acceptance-Aware Sampling (AAS) raises the mean resampling rounds, showing AAS improves sampling efficiency.

Figure 17 (left) shows the task weights assigned to Zebra throughout training. Our method initially assigns a lower weight to Zebra and gradually increases it as relative task performance evolves (see also Figure 6). In contrast, all baselines steadily reduce the weight assigned to Zebra over time. This adaptive reweighting explains the higher Zebra accuracy achieved by our method.

Figure 17 (right) reports the number of training steps required to reach specified worst-task accuracy thresholds in Experiment 2. Consistent with the trends observed in Figure 5, our method reaches all thresholds in fewer steps than the baselines, indicating faster progress on the weakest task. These results show that our approach not only improves final worst-task accuracy but also accelerates learning on under-performing tasks, even in the larger 9-task setting.

We further analyze the contribution of RP SAMPLER towards the performance of MT-GRPO. Figure 16 compares MT-GRPO with and without the ratio-preserving constraint (RPS). Without RPS, the method assigns higher weights to ARC to compensate for the mismatch between target task weights and the effective task representation in the training batch. However, this compensation is insufficient as the effective ARC representation remains lower than when RPS is enabled, highlighting the benefit of RP SAMPLER. Concretely, dropping the ratio-preserving constraint (while still filtering zero-gradient samples and resampling) leaves ARC with a *higher* mean assigned weight ($0.63$ vs. $0.43$) but a *lower* realized batch proportion ($0.36$ vs. $0.43$), and correspondingly lower ARC accuracy ($62.4\%$ vs. $64.8\%$). This shows that enforcing the ratio-preserving constraint is what converts the intended weight into actual gradient contribution.

Figure 16 (rightmost panel) compares our method with and without Acceptance-Aware Sampling (AAS), a key component of RP SAMPLER (see Algorithm 2). Removing AAS leads to a higher number of resampling rounds on average (the curve is smoothed for readability), demonstrating that AAS improves sampling efficiency.

### D.5. Computational Overhead and Wall-Clock Analysis

This section quantifies the per-step overhead introduced by the ratio-preserving sampler (Algorithm 2) and contrasts it with the resulting gain in wall-clock convergence, in the Experiment 1 setting.

**Per-step overhead:** MT-GRPO incurs additional cost due to the RP sampler, which resamples to maintain task ratios after filtering zero-gradient samples. For a fair comparison, we compare against DAPO (which also performs oversampling and

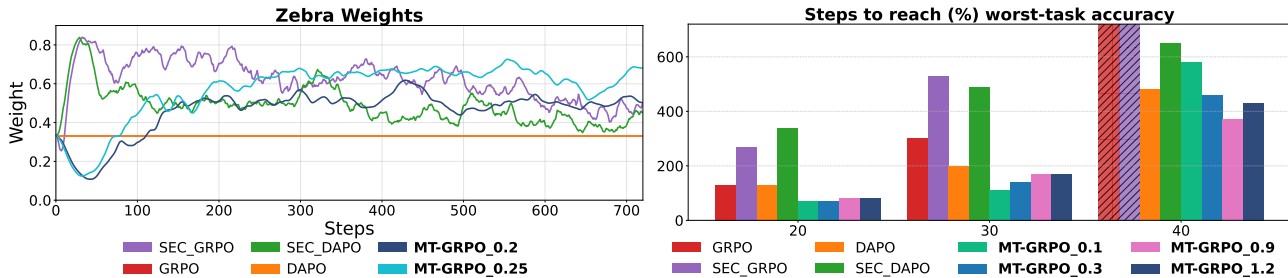

*Figure 17.* **Left:** Task weights assigned to Zebra across methods in Experiment 1. (See Figure 6 for full task performance and weights across other tasks.) **Right:** Number of training steps required to reach specified worst-task accuracy thresholds in Experiment 2. Bars reaching the maximum indicate that the method did not reach the threshold within the training budget. Our method consistently reaches thresholds in fewer steps than baselines.

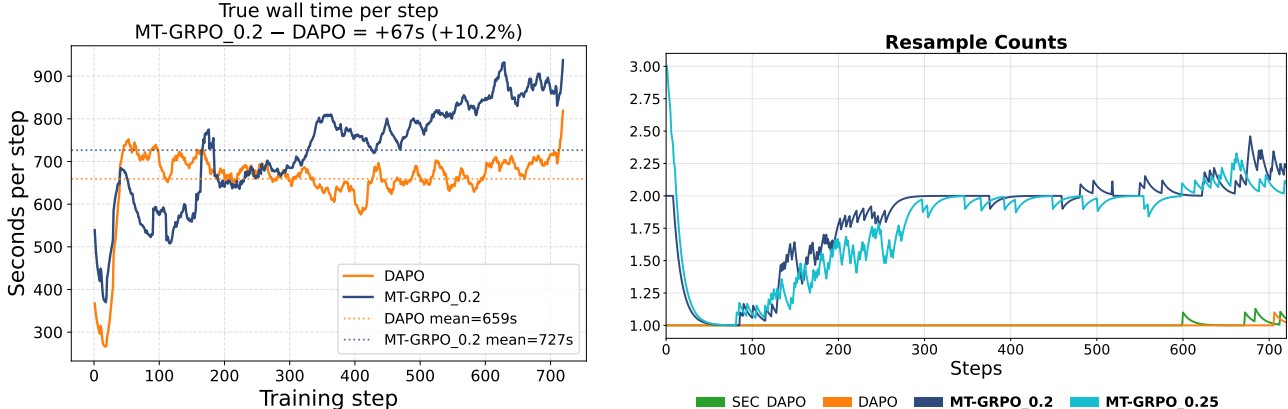

*Figure 18.* Per-step overhead in Experiment 1. **Left:** true wall time per step; MT-GRPO ($\lambda$=0.2) averages 727 s/step versus DAPO's 659 s/step (+67 s, +10.2%). **Right:** mean number of resampling rounds per step. DAPO uses a single generation pass throughout, while MT-GRPO issues up to ~2–2.25 rounds as filtering rates rise, which accounts for the per-step overhead.

filtering), rather than GRPO. In Experiment 1 (2×H200 GPUs), MT-GRPO is approximately +10.2% slower per step than DAPO (+67 s/step on average; Figure 18, left). This overhead is primarily due to additional resampling (Figure 18, right): since DAPO performs oversampling without enforcing task-wise ratios, it requires fewer regenerations than MT-GRPO.

**Wall-clock efficiency:** Despite this per-step overhead, MT-GRPO is more efficient overall. We evaluate all methods at a fixed 80-hour budget, which corresponds to the time required by GRPO to complete 720 steps in Experiment 1, ensuring a fair wall-clock comparison. At this budget (Table 1), MT-GRPO achieves significantly higher worst-task performance (by 10%) and reaches the target

*Table 1.* Wall-clock analysis (Experiment 1): Accuracy at 80-hour budget and time to reach specified worst-task thresholds.

| Method | Avg@80h | Worst@80h | 40% | 50% |
|---|---|---|---|---|
| GRPO | 53.7 | 34.5 | – | – |
| DAPO | 59.4 | 43.2 | 49.2 h | 98.0 h |
| MT-GRPO_0.2 | **60.6** | **53.6** | **17.6 h** | **38.6 h** |

thresholds (40%, 50%) much faster than both GRPO and DAPO, in spite of the per-step overhead. In particular, GRPO does not achieve 40% worst-task accuracy within this budget, and MT-GRPO reaches 50% in under 40 hours, compared to 98 hours for DAPO. Figure 19 shows the corresponding performance-versus-time plots.

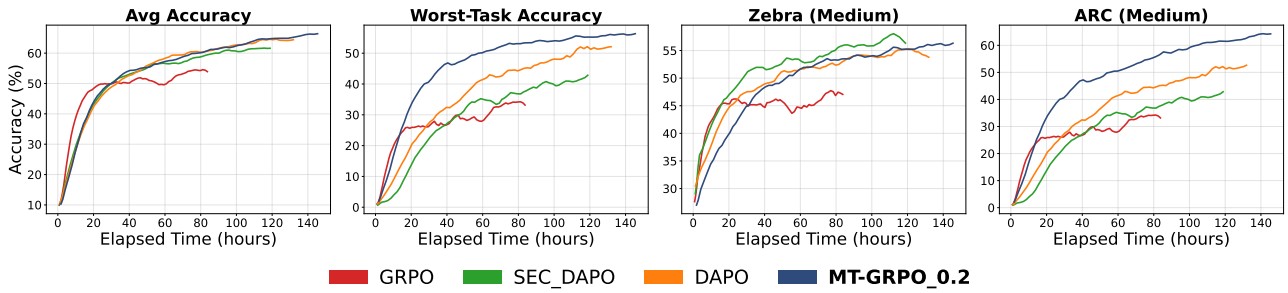

*Figure 19.* Accuracy versus elapsed wall-clock hours in Experiment 1. Even when measured in real time, MT-GRPO ($\lambda$=0.2) reaches target worst-task accuracy thresholds significantly faster than the baselines.

# E. Experimental Details and Reproducibility

For reproducibility, we provide our code at https://github.com/rsshyam/MT-GRPO.

We use the Qwen-2.5-3B base model for all experiments (Yang et al., 2024; Team, 2024). All methods are fine-tuned using the Volcano Engine Reinforcement Learning (verl) library (Sheng et al., 2025) for 720 training steps on two NVIDIA H200 (141GB) GPUs. We use the same versions of verl and relevant dependencies as in (Chen et al., 2025b) and incorporate the required DAPO modifications (Yu et al., 2025).

We use datasets released by Chen et al. (2025b), generated using the ReasoningGym framework (Stojanovski et al., 2025). The dataset contains easy, medium, and hard variants of Countdown, Zebra, and ARC. Each variant includes 1000 training instances and 200 test instances.

Rewards follow the protocol of Chen et al. (2025b): 1.0 for a correct answer, 0.1 for an incorrect answer with correct formatting, and 0 otherwise.

We estimate advantages using 72 rollouts for Experiment 1 and 8 rollouts for Experiment 2, generated using vLLM (Kwon et al., 2023) with temperature 1.0. We set the KL-divergence coefficient to 0, following standard practice in (Chen et al., 2025b; Yu et al., 2025). The maximum prompt length is 1024 tokens and the maximum response length is 4096 tokens.

We use the AdamW optimizer (Kingma, 2014) as implemented in verl, with learning rate $1e-6$ and betas (0.9, 0.99). For Experiment 1, we use a global batch size of 32 and PPO minibatch size of 8 due to the large number of rollouts. For Experiment 2, we use a global batch size of 256 and a minibatch size of 64. In verl, each global step consists of multiple minibatch updates and in our setup, each step performs four policy updates.

All methods use identical training configurations unless otherwise stated. For the SEC baseline (Chen et al., 2025b), we use the hyperparameters reported in the original paper. For both our method and DAPO, we apply the "clip higher" strategy from Yu et al. (2025) and set the clipping upper bound to 0.28.

**Metrics:** In addition to the worst-task accuracy and average accuracy, we have reported the *average relative change per task* metric, which quantifies how a method's performance differs from a reference baseline across tasks:

$$\Delta m\% = \frac{1}{K} \sum_{k=1}^{K} \frac{\text{Acc}_m(k) - \text{Acc}_b(k)}{\text{Acc}_b(k)} \times 100,$$

where $\text{Acc}_m(k)$ denotes the accuracy of method $m$ on task $k$, and $b$ denotes the reference baseline. For our analysis, we define the reference baseline to be DAPO. Positive values indicate average improvements over the baseline, while negative values indicate degradation. This metric is commonly used in multitask learning to account for differing task difficulty and accuracy scales (Liu et al., 2023; Navon et al., 2022). It complements worst-task accuracy by capturing whether improvements in robustness come at the expense of broader performance degradation.

**Filtering strategy:** For experiment 1, we enact a strict DAPO filtering strategy where the prompts with no rollouts with correct answer or all rollouts with correct answer are filtered out. For experiment 2, due to the increased task diversity and computational constraints, we adopt a more lenient filtering strategy following Yu et al. (2025) allowing prompts with non-constant rewards across rollouts (for e.g., due to correct/incorrect formatting).

**Hyperparameters for MT-GRPO:** For experiment 1, we use $\lambda = 0.2, 0.25$ and for experiment 2, we use $\lambda =$

---

**Algorithm 2** RATIO-PRESERVING SAMPLER (**RP SAMPLER**)

---

**Require:** Task weights $z \in \Delta^K$, batch size $B$, rollouts per prompt $N$, oversampling factor $M_{os}$, maximum resamples $N_{rs}$, tracked filtered ratios $\rho \in [0,1)^K$, maximum acceptance inflation factor $M_{acc}$

1: **Desired counts:** $(n_1, \ldots, n_K) \sim \text{Multinomial}(B, z)$

2: **Inflation factors:** $m_k \leftarrow \min \left\{ \frac{1}{1-\rho_k}, M_{acc} \right\} \quad \forall k$

3: **Recalibrated generation distribution:**

$$\hat{z}_k \leftarrow \frac{z_k \, m_k}{\sum_{j=1}^K z_j \, m_j}, \quad \forall k$$

4: **Sample:** $(\hat{n}_1, \ldots, \hat{n}_K) \sim \text{Multinomial}(M_{os}B, \hat{z})$

5: **for** $k = 1$ **to** $K$ **do**

6:     Sample prompts $\mathcal{P}_k$ from $D_k$ with $|\mathcal{P}_k| = \hat{n}_k$

7:     Generate rollouts $\mathcal{R}_k \leftarrow \{y_{i,j}^{(n)}\}_{j=1:\hat{n}_k, \, n=1:N}$

8: **end for**

9: $\mathcal{X} \leftarrow \text{ZERO\_GRAD\_FILTER}\left( \bigcup_k (\mathcal{P}_k, \mathcal{R}_k) \right)$

10: **Accepted counts:** $c_k \leftarrow |\{x \in \mathcal{X} : \text{task}(x) = k\}|$

11: **Deficiencies:** $\text{def}_k \leftarrow \max\{n_k - c_k, 0\}, \quad \forall k$

12: $r \leftarrow 1$    **(resampling round counter)**

13: **while** $\sum_{k=1}^K \text{def}_k > 0$ **and** $r \leq N_{rs}$ **do**

14:     **Deficiency-aware resampling distribution:**

$$\hat{z}_k^{(r)} \leftarrow \frac{\text{def}_k \, m_k}{\sum_{j=1}^K \text{def}_j \, m_j}, \quad \forall k$$

15:     $(a_1, \ldots, a_K) \sim \text{Multinomial}(M_{os}B, \, \hat{z}^{(r)})$

16:     **for** $k = 1$ **to** $K$ **do**

17:         **if** $a_k > 0$ **then**

18:             Sample $a_k$ new prompts $\tilde{\mathcal{P}}_k$ from $D_k$

19:             Generate $N$ rollouts $\tilde{\mathcal{R}}_k$ for each prompt

20:         **end if**

21:     **end for**

22:     $\tilde{\mathcal{X}} \leftarrow \text{ZERO\_GRAD\_FILTER}\left( \bigcup_k (\tilde{\mathcal{P}}_k, \tilde{\mathcal{R}}_k) \right)$

23:     $\mathcal{X} \leftarrow \mathcal{X} \cup \tilde{\mathcal{X}}$

24:     Update $c_k$ and $\text{def}_k \leftarrow \max\{n_k - c_k, 0\}$

25:     $r \leftarrow r + 1$

26: **end while**

27: **if** $|\mathcal{X}| \leq B$ **then**

28:     **Return** $\mathcal{X}$

29: **else**

30:     For each task $k$, retain at most $n_k$ samples from $\mathcal{X}_k$

31:     $\mathcal{X}^\star \leftarrow \bigcup_k \mathcal{X}_k^{(n_k)}$

32:     Sample $\mathcal{X}_r \subseteq \mathcal{X} \setminus \mathcal{X}^\star$ such that $|\mathcal{X}^\star \cup \mathcal{X}_r| = B$

33:     **Return** $\mathcal{X}^\star \cup \mathcal{X}_r$

34: **end if**

---

$0.1, 0.3, 0.9, 1.2$ for the trade-off parameter in Subroutine 1. We initialize the task weights $z^0$ uniformly and update them according to Subroutine 1 using current batch rewards $\{J_k(\theta_t)\}_{k=1}^K$, and improvements $\{I_k^{(t)}\}_{k=1}^K$ clipped to $\{0.1, 0.2\}$ for stability. We implement the gradient descent update in Line-5 of Subroutine 1 using AdamW (Kingma, 2014) with a learning rate of $0.025$ and weight decay $1e-5$ for experiment 1, and $1e-4$ for experiment 2. For the RP SAMPLER , we use an oversampling factor of $M_{os} = 3$, a maximum of $N_{rs} = 10$ resampling rounds for experiment 1, and $N_{rs} = 2$ for experiment 2, and inflate sampling probabilities based on filtering rates $\rho_k$ up to a maximum acceptance inflation factor $M_{acc}$ of $5$.

---

**Subroutine 2** Regularized Task-Weight Update

---

1: **Input:** rewards $\{J_k(\theta_t)\}_{k=1}^K$, logits $\xi_t$, stepsize $\beta$, regularization $\lambda$
2: $z_t \leftarrow \text{Softmax}(\xi_t)$
3: $(g_t)_k \leftarrow z_{k,t}\big(J_k(\theta_t) - \sum_{k=1}^K z_{k,t}J_k(\theta_t)\big)\ \forall k \in [K]$
4: $\xi_{t+1} \leftarrow \xi_t - \beta\big(g_t + \eta\xi_t\big)$
5: **Return:** $z_{t+1} = \text{Softmax}(\xi_{t+1})$

---

## F. Stabilizing Task-weight Updates through Regularization

We consider two practical mechanisms for improving the stability and effectiveness of task reweighting: (i) regularizing the task-weight dynamics, and (ii) incorporating task-level improvement signals. Here, we discuss the analysis of (i).

**Regularized task-weight update:** We add a $\ell_2$ regularization term to the *logit* update for $z = \text{Softmax}(\xi)$. This can be viewed as a practical mechanism for relaxing the strict worst-case behavior induced by the $\varepsilon = 0$ formulation and moving toward the smoother regime implied by $\varepsilon > 0$ in Equation (5).

Concretely, we update $\xi_t$ by gradient descent on $L(\xi_t)$ with an additional shrinkage term, yielding

$$\xi_{t+1} = \xi_t - \beta\big(g_t + \eta\xi_t\big), \tag{43}$$

where $g_t$ is given by

$$(g_t)_k = z_{k,t}\Big(J_k(\theta_t) - \sum_{j=1}^K z_{j,t}J_j(\theta_t)\Big). \tag{44}$$

The additional term $\eta\xi_t$ dampens the growth of logits and mitigates weight collapse. We summarize this regularized update in Subroutine 2.

**Is regularization sufficient?:** The regularized task-weight updates in Equation (43) mitigate rapid collapse of the weight distribution, but they continue to rely solely on the absolute task reward $J_k(\theta_t)$ as the signal for reweighting. However, absolute reward does not distinguish between tasks that are improving rapidly and tasks that have stagnated over the course of training. In multi-task post-training, this distinction is critical. A task that is currently weak but improving quickly may require less prioritization than a task with a similar reward that is no longer benefiting from updates. While increasing the regularization strength $\lambda$ in Subroutine 2 stabilizes the dynamics, it does so by driving the weights toward uniformity (average reward maximization), which weakens the prioritization of genuinely underperforming or slowly improving tasks (Désidéri, 2012; Yu et al., 2020; Liu et al., 2023). Consequently, even when collapse is avoided, absolute-reward-based reweighting alone may be insufficient to ensure that certain tasks do not remain under-optimized. These limitations motivate measuring how the loss (i.e., $-J_{\text{GRPO},k}(\theta)$) of each task evolves over training and introducing an improvement-aware task signal that explicitly captures how much each task benefits from policy updates, as done in Subroutine 1.

**Experiments:** We compare improvement-aware task reweighting (Subroutine 1) to regularized reward-only reweighting (Subroutine 2) in Experiments 1 and 2.

**Experiment 1 (3 tasks):** With weak regularization ($\eta = 1e-5$), the task-weight distribution collapses onto the current worst-performing task and largely ignores Countdown for most of training (see Figure 2), resulting in low worst-task and average accuracy. Stronger regularization ($\eta = 1e-2$) stabilizes training and yields similar worst-task accuracy but higher average accuracy and average relative change (Figure 20). These gains are driven primarily by improved performance on the easiest task (Countdown), whereas improvement-aware updates retain stronger performance on the harder task (ARC), which also exhibits high zero-gradient rates. This indicates that regularization tends to smooth weights toward uniformity rather than reassigning them to underperforming or slowly improving tasks.

**Experiment 2 (9 tasks):** Moderate regularization ($\eta = 5e-4$) improves worst-task accuracy over baselines, but remains below improvement-aware updates, particularly for $\lambda \in \{0.9, 1.2\}$ in terms of both worst-task accuracy and average relative change on hard tasks (Figure 21). Increasing the regularization to $1e-2$ degrades worst-task accuracy below DAPO. Although average accuracy increases, it remains below DAPO, and the gain is driven primarily by improvements on easy and medium tasks rather than hard tasks (see Figure 21, bottom).

In contrast, improvement-aware updates expose a controllable trade-off. Smaller values of the trade-off parameter (e.g., $\lambda \in \{0.1, 0.3\}$) prioritize average accuracy and progress on hard tasks (higher relative change) while maintaining worst-task accuracy competitive with baselines. Larger values of $\lambda$ focus primarily on improving worst-task performance.

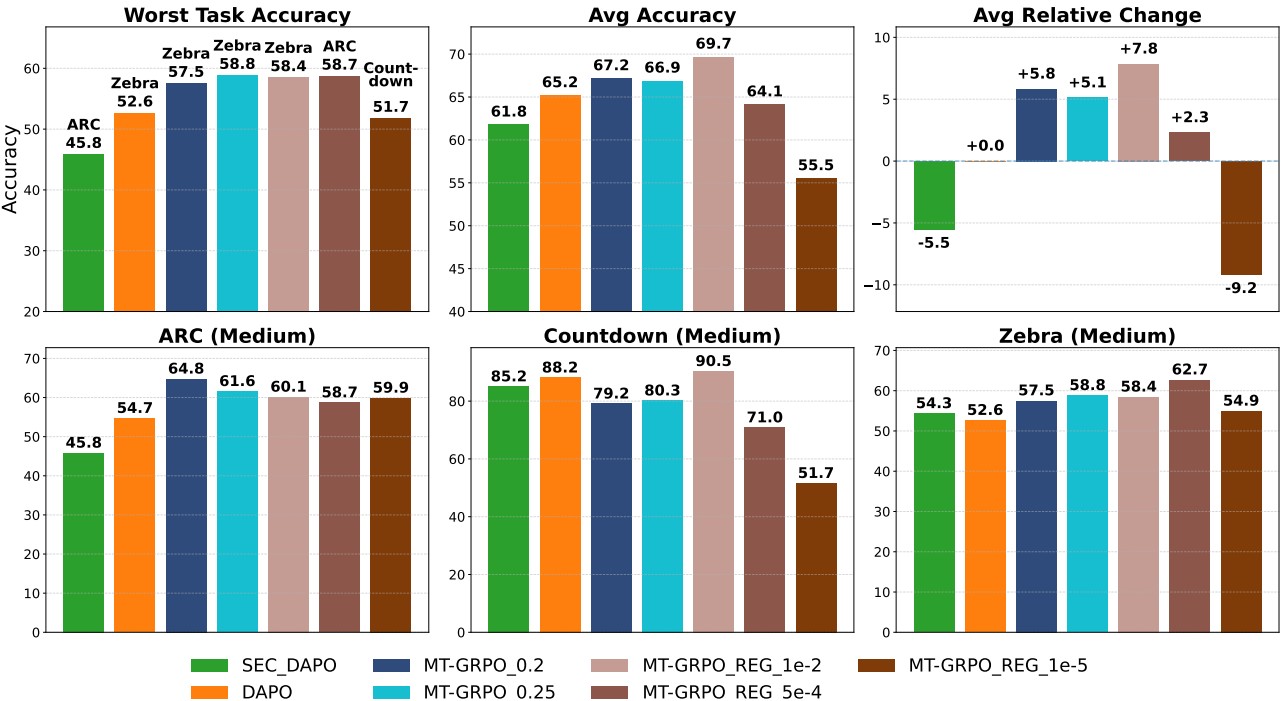

*Figure 20.* Experiment 1 (3 tasks): Comparison of improvement-aware updates (Subroutine 1) and regularized task-weight updates (Subroutine 2). With regularization $1e - 2$, the regularized task-weight update achieves similar worst-task accuracy but higher average accuracy, largely driven by gains on Countdown. However, improvement-aware updates achieve stronger performance on the harder task (ARC).

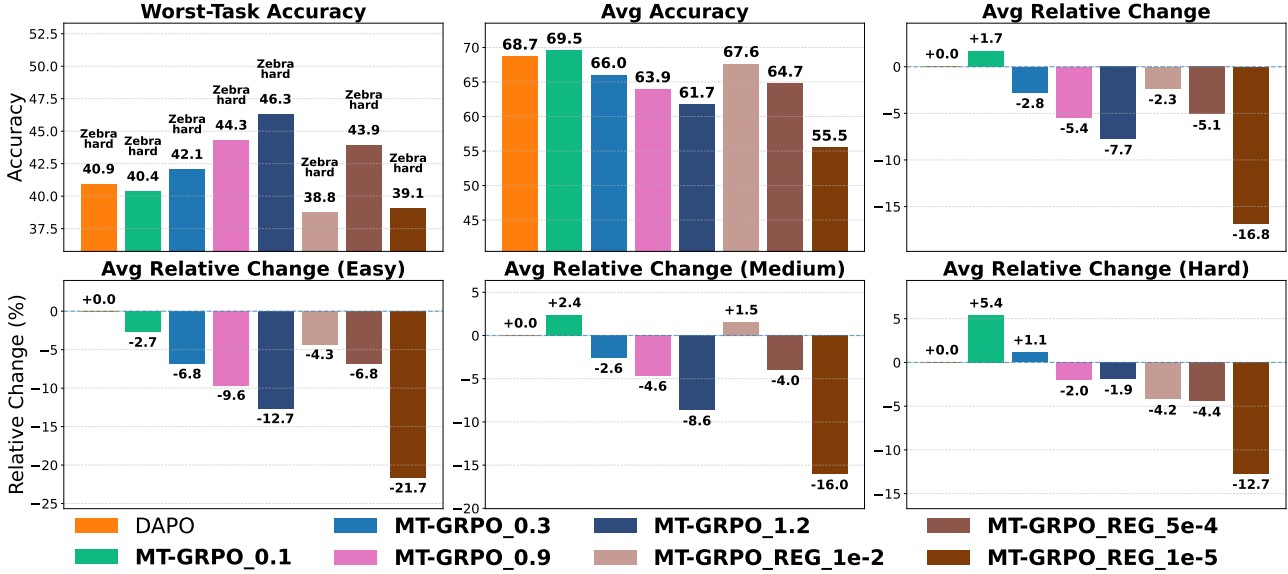

*Figure 21.* Experiment 2 (9 tasks): Comparison between improvement-aware updates (Subroutine 1) and regularized task-weight updates (Subroutine 2). Moderate regularization ($\eta = 5e - 4$) improves worst-task accuracy over baselines but underperforms improvement-aware updates on both worst-task accuracy and average relative change on hard tasks. Stronger regularization ($\eta = 1e - 2$) reduces worst-task accuracy below DAPO and yields only modest gains in average accuracy (still below DAPO), driven primarily by improvements on easy and medium tasks rather than hard tasks.

Overall, these results suggest that regularization can prevent weight collapse but is insufficient for reliably prioritizing under-optimized tasks. Incorporating improvement signals provides a more effective mechanism for allocating training emphasis to tasks that are under-performing or improving slowly.

