# OpenReview forum: "Multi-Task GRPO: Reliable LLM Reasoning Across Tasks"
_ICML.cc/2026/Conference — ICML 2026 regular_

### Official Review · Reviewer_WgS6 · 2026-03-10

**Soundness:** 3
**Presentation:** 3
**Significance:** 3
**Originality:** 3
**Overall Recommendation:** 4
**Confidence:** 3

**Summary:**

This paper studies the robustness of large language models in multi-task reinforcement learning post-training. The authors observe that existing RL post-training methods based on GRPO typically optimize either a single task or the average performance across tasks. In a multi-task training setting, however, such an objective often leads to imbalanced task performance. Easier tasks tend to dominate the training process, while harder tasks receive little improvement, which results in unstable capabilities across different reasoning tasks. To address this issue, the paper proposes Multi-Task GRPO (MT-GRPO), an RL post-training algorithm designed to improve multi-task reasoning ability. The proposed method consists of two key mechanisms: improvement-aware task reweighting and ratio-preserving sampling. Experiments are conducted on the Qwen-2.5-3B model with tasks including Countdown, Zebra puzzles, and ARC. The results show that, compared with baseline methods such as GRPO and DAPO, MT-GRPO achieves a substantial improvement in worst-task accuracy and requires fewer training steps to reach the target performance.

**Compliance With Llm Reviewing Policy:**

Affirmed.

**Final Justification:**

The responses have addressed my issues. I will keep the positive score.

**Key Questions For Authors:**

1. It would be helpful if the authors could clarify whether MT-GRPO has been evaluated on larger models, for example, models with at least 7B parameters. If such experiments have not yet been conducted, it would be valuable to discuss whether the authors expect the proposed method to remain stable when applied to larger-scale models.
2. The paper would also benefit from a more detailed analysis of component contributions. In particular, additional ablation experiments that separately evaluate the effects of task reweighting and the ratio-preserving sampling mechanism would help clarify how each component contributes to the overall performance gains.
3. It would also be useful to better understand the sensitivity of the method to the hyperparameter $\lambda$. Specifically, the authors could discuss whether the performance is stable across a reasonable range of $\lambda$ values and whether there is a recommended strategy for selecting this parameter when applying the method to new task sets.
4. Finally, it would be helpful to discuss the generality of the proposed method across different task types. The current experiments focus on structured reasoning tasks. It would therefore be interesting to know whether the method can also be applied to less structured settings, such as natural language question answering or code generation.

**Limitations:**

yes

**Strengths And Weaknesses:**

Strengths:
1. The method design is supported by a clear motivation. The paper identifies two key issues in multi-task GRPO training. First, optimizing the average reward leads to imbalanced performance across tasks. Second, the proportion of zero-gradient samples differs across tasks, which causes an imbalance in gradient contributions. The proposed mechanisms, improvement-aware task reweighting and ratio-preserving sampling, are specifically designed to address these two issues and therefore exhibit strong alignment with the underlying training dynamics.
2. The optimization objective is supported by a theoretical interpretation. The authors formulate the problem as a robust multi-task optimization objective with a max-min structure and relate it to prior work on distributionally robust optimization. This formulation provides a conceptual justification for focusing on the worst-performing tasks during training and offers a theoretical perspective on the proposed algorithm.
3. The paper also reports several useful observations about the training process. In particular, it notes that the GRPO loss does not always reflect the true difficulty of a task. For example, the loss may be identical when all generated outputs are correct or when all are incorrect. Based on this observation, the authors update task weights using task rewards rather than the GRPO loss. This design choice provides a practical insight into how task difficulty can be better captured during multi-task reinforcement learning.

Weaknesses:
1. The experimental scale is relatively limited. The experiments are conducted only on a 3B parameter model. In current large language model training, models are typically much larger. It therefore remains unclear whether the proposed method maintains its effectiveness on larger models such as 7B or beyond.
2. The diversity of task types is also limited. The evaluation focuses primarily on structured reasoning tasks, including Countdown, Zebra puzzles, and ARC. These tasks do not fully represent the diversity of tasks encountered in real-world applications, such as open-domain question answering, programming tasks, instruction following, and natural language inference. Validation on a broader range of task types would strengthen the empirical evidence of the paper.
3. The ablation study is insufficient. The proposed method contains several key components, including improvement-aware task weight updates and the ratio-preserving sampling mechanism. However, the paper does not provide sufficient ablation experiments to isolate and quantify the individual contribution of each component.

---

> ### Author Rebuttal · Authors · 2026-03-31
>
> We thank the reviewer for their positive assessment of our work, recognizing our clear motivation, interpretable theoretical formulation, and the insights into the GRPO training dynamics that underlie our method.
>
> **Scalability and generality of the proposed method**
>
> Our original evaluation uses structured reasoning tasks by design, providing a controlled setting where task difficulty, reward structure, and zero-gradient behavior can be precisely analyzed. This is important for isolating the core issue addressed in this work—imbalanced optimization dynamics in multi-task GRPO.
>
> We complement this with experiments on a larger model (OLMo-3 7B Instruct) and a more realistic, domain-diverse benchmark (SciKnowEval): https://anonymous.4open.science/r/icml-rebuttal-8DDE/sciknoweval-olmo3-7B/merged.png.
>
> SciKnowEval consists of natural-language QA tasks across biology, chemistry, physics, and materials science, and has been adopted in recent RL post-training work [1-4].
>
> Method|Avg|Worst
> ---|---|---
> SEC_GRPO|47.4|39.0
> GRPO|46.4|27.5
> GRPO_EQUAL|41.1|33.8
> SEC_DAPO|45.9|32.0
> DAPO|46.0|29.1
> DAPO_EQUAL|46.2|30.8
> MT-GRPO_0.5|**52.7**|**45.8**
>
> These results show that MT-GRPO **improves** worst-domain accuracy and yields more **balanced** optimization compared to baselines, matching trends in the controlled setting. Moreover, they demonstrate generalization to larger models and more realistic task mixtures, beyond structured reasoning settings.
>
> Since the method operates on reward signals and gradient allocation rather than task-specific structure, it is broadly applicable across diverse task types.
>
> **Ablation**
>
> We isolate the two components of MT-GRPO: **GRPO_IWU** (task reweighting) and **GRPO_RPS** (ratio-preserving sampling).
> GRPO_IWU replaces uniform sampling in GRPO with IWU-based weights (Subroutine 1), while GRPO_RPS keeps uniform weights but enforces task ratios in the post-filtered batch via resampling (Algorithm 2).
>
> Exp-1 (https://anonymous.4open.science/r/icml-rebuttal-8DDE/ablation/exp-1/fig_1.png):
>
> | Method | Avg | Worst |
> |---|---|---|
> | GRPO | 52.0 | 30.3 |
> | GRPO_IWU_0.25 | 43.9 | 41.9 |
> | DAPO | 65.2 | 52.6 |
> | GRPO_RPS | **68.9** | 54.6 |
> | MT-GRPO_0.25 | 66.9 | **58.8** |
>
> **Key observations:** IWU alone improves worst-task accuracy, confirming effective adaptive task reweighting, but suffers from a mismatch between assigned weights and effective gradient contributions due to task-dependent filtering (https://anonymous.4open.science/r/icml-rebuttal-8DDE/ablation/exp-1/grpo_iwu_arc.png):
>
> Method|Weight*|Batch Ratio*
> ---|---|---
> GRPO_IWU_0.25|0.7965|0.4487
>
> *- Mean over steps
>
> RPS eliminates this mismatch, ensuring batch proportions align with intended uniform weights, leading to highest average performance (https://anonymous.4open.science/r/icml-rebuttal-8DDE/ablation/exp-1/grpo_rps_batch_ratios.png ).
>
> Overall, IWU improves robustness but is unreliable without RPS, while RPS ensures correct gradient allocation but lacks adaptivity.
> MT-GRPO (IWU + RPS) **combines both**, achieving the best worst-task performance with competitive average accuracy.
>
> We plot the ablations for expt-2 in https://anonymous.4open.science/r/icml-rebuttal-8DDE/ablation/exp-2/fig4.png
>
> **Sensitivity to reward vs. improvement weighting (λ)**
>
> We refer the reviewer to Figure 7, where we evaluate MT-GRPO across a range of λ values. The results show a consistent and interpretable trade-off: increasing λ improves worst-task accuracy, while reducing average accuracy.
>
> Specifically, smaller λ (e.g., 0.1–0.3) emphasizes improvement signals, leading to larger gains on slower-learning (harder) tasks, at the cost of slight degradation on easier ones. In contrast, larger λ concentrates optimization on the worst-performing task, further improving worst-task accuracy but reducing overall average gains.
>
> Importantly, this behavior is stable and aligned with the objective in Eq. (5): λ explicitly controls the balance between worst-case optimization and average performance. For example, λ = 1.2 improves worst-task accuracy by up to +6% over DAPO, while λ = 0.1 yields the highest average per-task relative improvement.
>
> Overall, these results show that λ provides a practical and effective control knob for navigating the trade-off between worst-task robustness and average performance.
>
> References
>
> [1] Zhang et al., OpenRFT: Adapting Reasoning Foundation Model for Domain-Specific Tasks with Reinforcement Fine-Tuning, arXiv:2412.16849, 2024.
>
> [2] Xu et al., Not All Rollouts are Useful: Down-Sampling Rollouts in LLM Reinforcement Learning, arXiv:2504.13818, 2025.
>
> [3] He et al., Beyond Correctness: Confidence-Aware Reward Modeling for Enhancing Large Language Model Reasoning, EMNLP, 2025.
>
> [4] Hübotter et al., Reinforcement Learning via Self-Distillation, arXiv:2601.20802, 2026

---

> > ### Author Rebuttal · Reviewer_WgS6 · 2026-04-02
> >
> > Thank the authors for the detailed and thoughtful response. The explanations have fully addressed my concerns and clarified the points of uncertainty. I will maintain my evaluation as the weak accept.

---

> > > ### Author Response · Authors · 2026-04-08
> > >
> > > We thank the reviewer for maintaining their positive score following our detailed response and additional experiments.

---

### Official Review · Reviewer_ws9U · 2026-03-11

**Soundness:** 2
**Presentation:** 2
**Significance:** 3
**Originality:** 3
**Overall Recommendation:** 4
**Confidence:** 3

**Summary:**

This paper proposes a new algorithm, MT-GRPO, to address the problem of multi-task learning using GRPO. The objective of MT-GRPO involves worst-task reward maximiation and improvement-aware task reweighting. Thus, MT-GRPO can adapt the task weights dynamically at each step, and allocate more weights to tasks with lower rewards or slower improvement. Furthermore, to avoid the zero-gradient samples affecting the actual gradient contributions, this paper proposes RP Sampler that upsamples the rollouts. Experiments on 3-task and 9-task settings show that MT-GRPO outperforms baselines on worst-task accuracy, and achieves competitive average accuracy.

**Compliance With Llm Reviewing Policy:**

Affirmed.

**Final Justification:**

During the rebuttal phase, the authors provided extensive additional experiments that successfully address my previous concerns. In light of these improvements, I recommend this paper for acceptance.

**Key Questions For Authors:**

- Could the authors provide experimental results for a sequential training setting, where RL is applied to different tasks one after another? This simple baseline is commonly used in practice, and including such a comparison would help better demonstrate the advantages of the proposed method in realistic application scenarios.

- Does RP Sampler lead to more rollout cost? How much time does it slow down each step?

**Limitations:**

Research on RL algorithms typically relies on tasks with verifiable rewards.

**Strengths And Weaknesses:**

## Strengths
- This paper focuses on an important problem in post-training of LLMs, and is well motivated.
- The proposed algorithm MT-GRPO that formulates task reweighting as an optimization problem is theoretically sound.
- Experiments demonstrate that MT-GRPO can achieve multi-task robustness better and faster. It also outperforms baselines on average accuracy with proper hyper-parameters.

## Weaknesses

- The paper is not well structured. Some descriptions are rather redundant. For example, Section 2 can be more concise.

- While the use of synthetic and controlled datasets，specifically Countdown, Zebra Puzzles, and ARC, enables a precise and isolated evaluation of the algorithm’s mechanics, the paper is substantially weakened by the lack of experiments on more complex and realistic tasks, such as mathematics or code generation.

- The expansion to a 9-task setting is essentially a more granular decomposition of the original three taks into three distinct difficulty levels. While I accept this setup as a valid stress test for the algorithm's ability to handle varying zero-gradient rates within the same task family, it does not fully substitute for a true multi-task environment characterized by domain heterogeneity.

- There appears to be a lack of ablation studies for the key components of the proposed method, such as IWU and the RP Sampler. Figure 8 presents a comparison of weights with and without the RP Sampler, but it is unclear whether a corresponding comparison in terms of accuracy is provided. As the paper is quite lengthy, it is possible that I may have overlooked some results. If so, the authors could clarify where these ablations are reported. Otherwise, including explicit ablation studies would be important to better understand the contribution of each component.

---

> ### Author Rebuttal · Authors · 2026-03-31
>
> We thank the reviewer for recognizing the importance of the problem, the principled formulation of MT-GRPO, and its strong empirical performance.
>
> **Scalability and broader task mixtures**
>
> Our original evaluation uses structured reasoning tasks and provides a controlled setting for isolating the core issue addressed in this work—imbalanced optimization dynamics in multi-task GRPO. We complement this with experiments on a larger model (OLMo-3 7B Instruct) and a more realistic, domain-diverse benchmark (SciKnowEval):
>  https://anonymous.4open.science/r/icml-rebuttal-8DDE/sciknoweval-olmo3-7B/merged.png .
>
> SciKnowEval consists of natural-language QA tasks across biology, chemistry, physics, and materials science, and has been adopted in recent RL post-training works [1-4].
>
> Method|Avg|Worst
> ---|---|---
> SEC_GRPO|47.4|39.0
> GRPO|46.4|27.5
> GRPO_EQUAL|41.1|33.8
> SEC_DAPO|45.9|32.0
> DAPO|46.0|29.1
> DAPO_EQUAL|46.2|30.8
> MT-GRPO_0.5|**52.7**|**45.8**
>
> These results show that MT-GRPO **improves** worst-domain accuracy and yields more **balanced** optimization compared to baselines. This matches trends in the controlled setting and demonstrates generalization to larger models and more realistic task mixtures.
>
> **Ablation**
>
> We isolate the two components of MT-GRPO: **GRPO_IWU** (task reweighting) and **GRPO_RPS** (ratio-preserving sampling). GRPO_IWU replaces uniform sampling in GRPO with IWU-based weights (Subroutine 1), while GRPO_RPS keeps uniform weights but enforces task ratios in the post-filtered batch via resampling (Algorithm 2).
>
> Exp-1 (https://anonymous.4open.science/r/icml-rebuttal-8DDE/ablation/exp-1/fig_1.png):
>
> | Method | Avg | Worst |
> |---|---|---|
> | GRPO | 52.0 | 30.3 |
> | GRPO_IWU_0.25 | 43.9 | 41.9 |
> | DAPO | 65.2 | 52.6 |
> | GRPO_RPS | **68.9** | 54.6 |
> | MT-GRPO_0.25 | 66.9 | **58.8** |
>
> **Key observations:** IWU alone improves worst-task accuracy, confirming effective adaptive task reweighting, but suffers from a mismatch between assigned weights and effective gradient contributions due to task-dependent filtering (https://anonymous.4open.science/r/icml-rebuttal-8DDE/ablation/exp-1/grpo_iwu_arc.png):
>
> Method|Weight*|Batch Ratio*
> ---|---|---
> GRPO_IWU_0.25|0.7965|0.4487
>
> RPS eliminates this mismatch, ensuring batch proportions align with intended uniform weights, leading to highest average performance.
> (https://anonymous.4open.science/r/icml-rebuttal-8DDE/ablation/exp-1/grpo_rps_batch_ratios.png)
>
> Overall, IWU improves robustness but is unreliable without RPS, while RPS ensures correct gradient allocation but lacks adaptivity. MT-GRPO (IWU + RPS) **combines both**, achieving the best worst-task performance with competitive average accuracy.
>
> We plot the ablations for expt-2 in https://anonymous.4open.science/r/icml-rebuttal-8DDE/ablation/exp-2/fig4.png
>
> **Figure 8 Clarification**
>
> Figure 8 focuses on weight vs. batch-ratio alignment for ARC when MT-GRPO is run without the ratio-preserving constraint, while retaining filtering of non-zero gradient samples and resampling. We observe that, despite receiving a higher assigned weight, ARC has lower realized batch proportions without the ratio-preserving constraint, leading to lower ARC performance (https://anonymous.4open.science/r/icml-rebuttal-8DDE/arc-fig8-clar/arc_focus_1x3.png).
>
> Method|Weight*|Batch Ratio*|Acc
> ---|---|---|---
> MT-GRPO_0.2|0.4311|0.4337|64.75
> MT-GRPO_0.2_NO_RPS|0.6343|0.3586|62.44
>
> We will revise the paper to include these accuracy comparisons alongside Figure 8 for clarity.
>
> **Per-step overhead**
>
> MT-GRPO incurs additional cost due to resampling in the RP sampler. In Exp-1 (2×H200 GPUs), it is approximately +10.2% slower per step than DAPO (+67s/step):
> https://anonymous.4open.science/r/icml-rebuttal-8DDE/overhead/exp-1/wall_time.png
>
> Despite this, MT-GRPO is **more efficient overall**. At a fixed 80-hour budget (≈720 GRPO steps), it achieves significantly higher worst-task performance and reaches target performance thresholds **much faster**.
>
> Method|Worst@80h|50%
> ---|---|---
> GRPO|34.5|–
> DAPO|43.2|98.0h
> MT-GRPO_0.2|53.6|38.6h
>
> We kindly refer the reviewer to our response to Reviewer ncty for a more comprehensive runtime analysis.
>
> **Sequential Baseline**
>
> Sequential training is an interesting and practically relevant setting. However, it addresses a different problem from the one considered in this work, which focuses on joint multi-task training. Sequential training introduces additional challenges such as catastrophic forgetting and sensitivity to task ordering, and typically requires evaluation protocols from the continual learning literature (e.g., [6]). We therefore consider it beyond the scope of this work. We are happy to discuss this further with the reviewer if they disagree.
>
> *- Mean over steps
>
> [6] Li et al., "Omni-Thinker: Scaling Multi-Task RL in LLMs with Hybrid Reward and Task Scheduling." arXiv:2507.14783 (2025).

---

> > ### Author Rebuttal · Reviewer_ws9U · 2026-04-01
> >
> > Thank you for the authors’ detailed and constructive response. The newly added experiments on broader task mixtures and ablations have addressed part of my concerns, and I have therefore increased my score from 3 to 4.
> >
> > However, my concerns regarding the paper’s organization and writing have not been addressed. This issue caused noticeable difficulty during my reviewing.
> >
> > Regarding the Sequential Baseline, I would like to reiterate my motivation for suggesting this comparison. In many practical LLM training pipelines that I am familiar with, when handling multi-task RL scenarios, the simple sequential training strategy serves as a fast, easily deployable, and often acceptable solution. Therefore, including such a baseline would make the paper more practically relevant and strengthen its impact.
> >
> > Importantly, the Sequential Baseline does not require modifications to the existing training framework, making it particularly attractive for industry adoption. In contrast, the proposed method introduces additional engineering overhead for integration. As a result, for the method to be widely adopted in practice, it needs to demonstrate a clear and substantial advantage over such simple baselines. Marginal improvements may not be sufficient to justify the additional implementation cost.
> >
> > I recommend that the authors consider including this comparison in the next version of the paper, as it would significantly improve the paper’s practical value and facilitate broader adoption.

---

> > > ### Author Response · Authors · 2026-04-07
> > >
> > > We thank the reviewer for their detailed response and for increasing their score following our additional experiments and ablations.
> > >
> > > ## Presentation and Organization
> > >
> > > Thank you for emphasizing the paper’s organization. We take this concern seriously and intend to improve this aspect of the paper. In the revision, we will streamline Sections 1-2 by presenting the limitations of prior work and the motivation more concisely, with a sharper focus on the core ideas of the paper. We will also move some standard technical details from later sections to the appendix, and make targeted edits throughout to improve clarity and reduce redundancy. We believe these changes will improve readability, and we welcome any additional suggestions from the reviewer.
> > >
> > > ## Sequential Baseline
> > >
> > > In response to the reviewer’s follow-up comment on the practical relevance of sequential training as a simple, deployable baseline, we conducted additional experiments to evaluate this in the Experiment 2 setup (9 tasks) under a matched total budget (720 steps).
> > >
> > >
> > > We initially considered a fine-grained 9-stage schedule across individual difficulty levels (easy/medium/hard) for each task; however, in practice, earlier stages (e.g., Countdown easy) quickly saturated, leading to a large proportion of zero-gradient/all-correct samples. This led to significant inefficiency for DAPO and instability for GRPO during training.
> > >
> > > To provide a more practical and representative baseline, we adopt a family-level sequential setup, merging all difficulty levels within each task and training sequentially over Countdown, Zebra, and ARC for 240 steps each. We evaluate multiple task orders to account for ordering effects (e.g., Countdown→Zebra→ARC (CZA), Zebra→ARC→Countdown (ZAC)), and consider both DAPO and GRPO.
> > >
> > >
> > > | Method       |  Avg Acc (%) | Worst-Task (%) |
> > > |-----------------|---------------|----------------|
> > > | GRPO         | 56.0        | 30.4           |
> > > | SEQ-GRPO (CZA) | 58.0        | 37.9           |
> > > | SEQ-GRPO (ZAC) | 61.9        | 41.1           |
> > > | DAPO         |  68.7        | 40.9           |
> > > | SEQ-DAPO (CZA)   | 63.7        | 39.4           |
> > > | SEQ-DAPO (ZAC)   | 65.0        | 36.6           |
> > > | MT-GRPO_0.1  | **69.5**        | 40.4           |
> > > | MT-GRPO_1.2  | 61.7        | **46.3**           |
> > >
> > >
> > > Across these configurations, we observe a consistent pattern: sequential training underperforms on worst-task accuracy compared to MT-GRPO across both GRPO and DAPO variants. Even in terms of average accuracy, sequential variants remain below joint DAPO training with uniform sampling (see: <https://anonymous.4open.science/r/icml-rebuttal-8DDE/seq/compare_2x3_seq_all.pdf>).
> > >
> > > This is because sequential training does not explicitly account for task-wise robustness or reallocate optimization effort toward weaker or slowly improving tasks. In contrast, MT-GRPO jointly optimizes all tasks while dynamically reweighting them based on reward and improvement signals, and ensures correct gradient allocation via the RP sampler, resulting in more stable and **robust** performance across tasks.
> > >
> > > Moreover, we observe clear signs of forgetting, where performance on earlier tasks degrades after switching to later ones—particularly for Countdown and ARC (see: <https://anonymous.4open.science/r/icml-rebuttal-8DDE/seq/forget_1x3_hard_seq_lines.pdf>). We also observe that different task orders lead to noticeably different final average and worst-task accuracies, highlighting sensitivity to task ordering and the non-triviality of identifying an effective ordering, especially as the number of tasks grows.
> > >
> > >
> > > We will include the additional experimental results discussed during the rebuttal in the revised version.

---

### Official Review · Reviewer_ncty · 2026-03-13

**Soundness:** 3
**Presentation:** 2
**Significance:** 3
**Originality:** 2
**Overall Recommendation:** 4
**Confidence:** 3

**Summary:**

This paper studies multi-task reinforcement learning post-training with GRPO for large language models. The authors observe that when GRPO is applied to multiple tasks simultaneously, optimization can become imbalanced: easier tasks tend to dominate training while harder tasks receive limited improvement. In addition, GRPO can produce zero-gradient prompts when sampled responses receive identical rewards, and the frequency of such cases varies across tasks, leading to discrepancies between intended task weights and effective gradient contributions.

To address these issues, the paper proposes Multi-Task GRPO (MT-GRPO). The approach introduces (1) an improvement-aware task reweighting mechanism that dynamically adjusts task weights based on reward and improvement signals, and (2) a ratio-preserving sampler that maintains desired task proportions after filtering zero-gradient samples.

**Compliance With Llm Reviewing Policy:**

Affirmed.

**Key Questions For Authors:**

（1）Have the authors evaluated MT-GRPO on larger models or more realistic training settings?

（2）How does the method perform on more heterogeneous task mixtures beyond reasoning tasks?

（3）What is the additional computational cost introduced by the ratio-preserving sampler?

（4）How sensitive is performance to the weighting between reward-based and improvement-based signals?

（5）Does improving worst-task performance during training translate to improved robustness on unseen tasks?

（6）Could the code be made public anonymously?

**Limitations:**

The paper evaluates the proposed approach on a relatively small model and a limited set of reasoning tasks, leaving open questions about scalability and applicability to broader multi-task settings. In addition, the computational overhead introduced by the ratio-preserving sampling mechanism is not thoroughly analyzed.

**Strengths And Weaknesses:**

Strengths：

（1）The paper identifies an important issue in multi-task RL post-training, namely the imbalance of optimization across tasks.

（2）The analysis of GRPO dynamics, particularly the role of zero-gradient prompts and task-dependent gradient contributions, is useful and helps motivate the proposed solution.

（3）The proposed components are relatively simple and can be incorporated into existing GRPO-based pipelines with minimal modification.

（4）The experiments show gains in worst-task accuracy and improved convergence for weaker tasks.

Weakness：

（1）The proposed techniques (adaptive task weighting and constrained sampling) resemble existing ideas in curriculum learning and robust optimization. The main contribution lies in adapting these ideas to GRPO-based RL post-training.

（2）Experiments are conducted on a single relatively small model (Qwen-2.5-3B) and a small set of reasoning tasks. It is unclear whether the approach scales to larger models or more diverse task mixtures.

（3）The evaluation mainly focuses on worst-task accuracy and convergence dynamics. Additional evaluations on broader task mixtures or downstream generalization would strengthen the claims.

（4）The RP sampler introduces additional sampling constraints, but the computational overhead compared to standard GRPO is not clearly reported.

---

> ### Author Rebuttal · Authors · 2026-03-30
>
> We thank the reviewer for their positive assessment of our work, recognizing the importance of the problem, our analysis of GRPO dynamics, and the practicality of our approach.
>
> **Novelty**
>
> We agree that, at a high level, adaptive weighting and constrained sampling relate to prior work on curriculum learning and imbalance-aware training. That said, we believe the main novelty of our work is the GRPO-specific formulation of the multi-task imbalance problem and its correction. MT-GRPO is designed specifically to address this imbalance: (i) the improvement-aware reweighting is defined around task performance and progress under GRPO updates, rather than a generic curriculum heuristic, and (ii) the ratio-preserving sampler is introduced to maintain the desired task proportions after zero-gradient filtering, which is a GRPO-specific failure mode not handled by standard multi-task weighting schemes. We therefore view the contribution as a principled adaptation to RL post-training with GRPO, supported by analysis and empirical gains in weak-task and worst-case performance. We will make this distinction clearer in the revision by better positioning prior work and by emphasizing our contribution.
>
> **Scalability and broader task mixtures**
>
> Our original evaluation uses structured reasoning tasks and provides a controlled setting for isolating the imbalanced optimization dynamics in multi-task GRPO. We complement this with experiments on a larger model (OLMo-3 7B Instruct) and a more realistic, domain-diverse benchmark (SciKnowEval):
>  https://anonymous.4open.science/r/icml-rebuttal-8DDE/sciknoweval-olmo3-7B/merged.png .
>
> SciKnowEval consists of natural-language QA tasks across biology, chemistry, physics, and materials science, and has been adopted in recent RL post-training works [1-4].
>
> Method|Avg|Worst
> ---|---|---
> DAPO|46.0|29.1
> SEC_GRPO|47.4|39.0
> MT-GRPO_0.5|52.7|**45.8**
>
> These results show that MT-GRPO **improves** worst-domain performance and demonstrates generalization to larger models and more realistic task mixtures (see our response to Reviewer 8img for additional baselines).
>
> **Per step overhead**
>
> MT-GRPO incurs additional cost due to RP sampler, which resamples to maintain task ratios after filtering zero-gradient samples. For a fair comparison, we compare against DAPO (which also performs oversampling, filtering), rather than GRPO.
> In Exp-1 (2×H200 GPUs), MT-GRPO is approximately +10.2% slower per step than DAPO (+67s/step on average):
> https://anonymous.4open.science/r/icml-rebuttal-8DDE/overhead/exp-1/wall_time.png
>
> This overhead is primarily due to additional resampling ( https://anonymous.4open.science/r/icml-rebuttal-8DDE/overhead/exp-1/regen.pdf ). Since DAPO performs oversampling without enforcing task-wise ratios, it requires fewer regenerations than MT-GRPO.
>
> Despite this per-step overhead, MT-GRPO is **more efficient** overall. We evaluate all methods at a fixed 80-hour budget, which corresponds to the time required by GRPO to complete 720 steps in Exp-1, ensuring a fair wall-clock comparison.
> At this budget, MT-GRPO achieves significantly higher worst-task performance (by 10%) and reaches target thresholds (40%,50%) **much faster** than both GRPO and DAPO in spite of per-step overhead:
>
> Method|Avg@80h|Worst@80h|40%|50%
> ---|---|---|---|---
> GRPO|53.7|34.5|–|–
> DAPO|59.4|43.2|49.2h|98.0h
> MT-GRPO_0.2|**60.6**|**53.6**|**17.6h**|**38.6h**
>
> In particular, GRPO does not achieve 40% worst-task accuracy within this budget. Moreover, MT-GRPO reaches 50% in under 40 hours, compared to 98 hours for DAPO. We will include explicit runtime comparisons and performance-vs-time plots (https://anonymous.4open.science/r/icml-rebuttal-8DDE/overhead/exp-1/fig1_steps_time.pdf) in the revised version.
>
> **Robustness to unseen tasks**
>
> Our current evaluation focuses on robustness within the training task mixture, where MT-GRPO improves worst-task performance and yields more balanced optimization. Improved robustness on unseen tasks typically requires a different training strategy, since it involves optimizing the training mixture w.r.t. held-out target tasks. This is consistent with recent data-mixture optimization work [5], which typically uses target-task validation signals (e.g., validation loss or gradient alignment) to guide mixture selection for downstream targets. We therefore view unseen-task robustness as an important but distinct direction for future work.
>
> **Code availability**
>
> An anonymous code repository is already included in the submission (see line 1148).
>
> **Sensitivity to reward vs. improvement weighting (λ)**
>
> We evaluate MT-GRPO for a range of λ values in Figure 7. Increasing λ improves worst-task accuracy while reducing average accuracy, providing a controlled trade-off aligned with Eq. (5). We kindly refer the reviewer to our response to Reviewer WgS6 for a detailed discussion.
>
>
> [5] Fan et al., "GRAPE: Optimize Data Mixture for Group Robust Multi-target Adaptive Pretraining." Neurips 2025

---

> > ### Author Rebuttal · Reviewer_ncty · 2026-04-03
> >
> > I appreciate the authors’ response. However, the new concerns raised remain unresolved, I will not be able to maintain my positive rating.

---

> > > ### Author Response · Authors · 2026-04-05
> > >
> > > We thank the reviewer again for their feedback.
> > >
> > > We addressed the reviewer’s main concerns in the rebuttal as follows: (i) scalability and broader task mixtures via additional experiments on **OLMo-3 7B and SciKnowEval**, (ii) computational overhead via **quantitative runtime analysis**, and (iii) sensitivity and robustness through additional discussion and analysis.
> > >
> > > Regarding runtime analysis, we note that on OLMo3–SciKnowEval (150 steps), MT-GRPO achieves **much higher** peak worst-task performance than all baselines, despite reaching it later. Moreover, baselines significantly degrade after their peak, unlike MT-GRPO (see: https://anonymous.4open.science/r/icml-rebuttal-8DDE/overhead/sciknow/sciknow_only_lines_time.pdf). We could not include these results earlier due to space constraints.
> > >
> > > To further address scalability and task diversity, we extended our evaluation during the discussion period to another **7B** model (Qwen2.5-7B) and a more **diverse multi-task** setting combining **MATH** (Hendrycks et al., 2021), **ARC**, and **SciKnowEval (Chemistry, Physics)**. This setup spans heterogeneous task types, including mathematical reasoning, inductive reasoning (ARC), and natural language QA.
> > >
> > > We train MT-GRPO with λ ∈ {0.05, 0.2} and the primary baselines from the paper (GRPO, DAPO, SEC-GRPO, SEC-DAPO) for 150 steps.
> > >
> > > MT-GRPO **substantially improves** worst-task performance (ARC) and yields more balanced optimization compared to all baselines (**37.6 vs 25.0** for DAPO). Moreover, larger λ improves worst-task accuracy at the cost of average accuracy, consistent with the trade-off discussed in our experimental section (see Figure 7, Section 6.2).
> > >
> > > The results are shown below (see https://anonymous.4open.science/r/icml-rebuttal-8DDE/multi-domain/qwen7B_merged_2x3.png ):
> > >
> > > | Name          | Avg Acc (%) | Worst-Task Acc (%) |
> > > |---------------|------------:|-------------------:|
> > > | SEC_GRPO      | 52.6        | 8.9                |
> > > | GRPO          | 55.4        | 15.0               |
> > > | SEC_DAPO      | 56.1        | 8.8                |
> > > | DAPO          | 58.3        | 25.0               |
> > > | MT-GRPO_0.05   |    56.2     |        33.1        |
> > > | MT-GRPO_0.2   | 51.8        | **37.6**               |
> > >
> > > Runtime analysis further shows that MT-GRPO reaches target worst-task accuracy thresholds **significantly faster** than baselines. (see: https://anonymous.4open.science/r/icml-rebuttal-8DDE/multi-domain/qwen7B_lines_time.png )
> > >
> > > These additional experiments further **strengthen** our findings by demonstrating that MT-GRPO generalizes to larger models and heterogeneous, realistic multi-task settings.
> > >
> > > Given this, we believe that we have fully addressed the concerns raised in the original review.

---

### Official Review · Reviewer_8img · 2026-03-13

**Soundness:** 2
**Presentation:** 3
**Significance:** 3
**Originality:** 3
**Overall Recommendation:** 5
**Confidence:** 4

**Summary:**

This paper proposes a variant of GRPO for multi-task scenarios. The key insight is that the optimal task weights for effective learning are dynamic throughout the training process. Simply filtering samples based on reward variance does not reliably capture this dynamic. Instead, the paper proposes a new task data sampling strategy that accounts for both task performance and step-wise improvement.

**Compliance With Llm Reviewing Policy:**

Affirmed.

**Final Justification:**

This paper addresses an under-explored, practical problem using an intuitive method. While the scale of the experiments remains a concern, the paper's strengths outweigh its weaknesses. I recommend acceptance.

**Key Questions For Authors:**

See above.

**Limitations:**

This paper should add more discussion on its limitations.

**Strengths And Weaknesses:**

* Soundness: The key contribution (an improved multi-task data sampler) is empirically validated through controlled experiments on a dedicated setup involving three tasks. The performance improvements across multiple perspectives are evident. However, there is a concern regarding the scale of the experiments, which leaves the generalizability of the conclusions unclear. Although the paper provides two sets of experiments, they are conducted using the same group of tasks and the same base model. Furthermore, the baseline for fixed task weights only considers uniform sampling. The performance gap between the proposed method and a more optimized fixed task weight remains unclear.

* Presentation: The paper is well-presented and grounded in the existing literature.

* Significance: This work presents a simple yet effective technique with intuitive motivation to address a practical problem in general RL for LLMs. Although the empirical results are not entirely comprehensive, the insights provided are beneficial to the research community.

* Originality: The paper provides new insights into multi-task RL for LLMs. The contribution is clearly demonstrated, building effectively upon GRPO and dynamic data sampling for RL.

---

> ### Author Rebuttal · Authors · 2026-03-30
>
> We thank the reviewer for their positive assessment of our work, recognizing the intuitive motivation of our approach, our improved multi-task sampling strategy, and its empirical validation.
>
> **Scalability and broader task mixtures**
>
> Our original evaluation uses structured reasoning tasks (Countdown, Zebra, ARC) by design, providing a controlled setting where task difficulty, reward structure, and zero-gradient behavior can be precisely analyzed. This is important for isolating the core issue addressed in this work—imbalanced optimization dynamics in multi-task GRPO. We complement this with experiments on a larger model (OLMo-3 7B Instruct) and a more realistic, domain-diverse benchmark (SciKnowEval): https://anonymous.4open.science/r/icml-rebuttal-8DDE/sciknoweval-olmo3-7B/merged.png.
>
> SciKnowEval consists of natural-language QA tasks across biology, chemistry, physics, and materials science, and has been adopted in recent RL post-training work [1–4].
>
> | Method | Avg Acc (%) | Worst-Task (%) |
> | :---- | :---- | :---- |
> | SEC\_GRPO | 47.4 | 39.0 |
> | GRPO | 46.4 | 27.5 |
> | GRPO\_EQUAL | 41.1 | 33.8 |
> | SEC\_DAPO | 45.9 | 32.0 |
> | DAPO | 46.0 | 29.1 |
> | DAPO\_EQUAL | 46.2 | 30.8 |
> | MT-GRPO\_0.5 | **52.7** | **45.8** |
>
> These results show that MT-GRPO **improves** worst-domain accuracy and yields more **balanced** optimization compared to baselines. This matches trends in the controlled setting and demonstrates generalization to larger models and more realistic task mixtures.
>
> **Stronger fixed-weight baselines**
>
> We note that any a priori fixed weighting can inevitably be mismatched across different phases of training. Our uniform-sampling baseline (DAPO) already illustrates this issue: task learning progress is highly imbalanced and evolves substantially over training. For example, task accuracies change from approximately (Zebra 30.4, ARC 1.2, Countdown 1.1) early in training, to (52.6, 41.6, 85.4) mid-training, and finally to (52.6, 54.7, 88.2) (see Figure-6: top plots in paper, https://anonymous.4open.science/r/icml-rebuttal-8DDE/Task-weights/medium_random_dapo_checkpoint_bars.png and table below).
>
> | Task | Beginning | Mid | Final |
> | :---- | :---- | :---- | :---- |
> | ARC | 1.2 | 41.6 | 54.7 |
> | Countdown | 1.1 | 85.4 | 88.2 |
> | Zebra | 30.4 | 51.7 | 52.6 |
>
> This progression highlights that the relative need for optimization shifts over time. Early in training, ARC and Countdown are the clear bottlenecks; a reasonable fixed strategy might therefore prioritize them. However, by the middle stage, Countdown has already advanced significantly (0.85), so further prioritizing it would allocate resources to a task that is no longer limiting overall performance. Conversely, focusing on ARC based on its mid-stage gap would neglect Zebra, which remains comparatively weaker at later stages. Similarly, ignoring Zebra early due to its relatively higher initial accuracy leads to suboptimal final performance, as it becomes a bottleneck later.
>
> In principle, one could tune fixed weights based on final outcomes and retrain from scratch, but this relies on **hindsight about training dynamics** and is not a practical strategy. Moreover, such choices depend on the training horizon, as the relative task imbalance—and thus the “optimal” weighting—changes depending on when training is stopped.
>
> MT-GRPO addresses this directly by adapting task allocation online based on current performance and improvement signals, enabling it to respond to shifting bottlenecks without relying on fixed or hindsight-driven weight selection.
>
> **Limitations** We will expand the discussion of limitations in the revised version.
>
> References
>
> [1] Zhang et al., OpenRFT: Adapting Reasoning Foundation Model for Domain-Specific Tasks with Reinforcement Fine-Tuning, arXiv:2412.16849, 2024.
>
> [2] Xu et al., Not All Rollouts are Useful: Down-Sampling Rollouts in LLM Reinforcement Learning, arXiv:2504.13818, 2025.
>
> [3] He et al., Beyond Correctness: Confidence-Aware Reward Modeling for Enhancing Large Language Model Reasoning, EMNLP, 2025.
>
> [4] Hübotter et al., Reinforcement Learning via Self-Distillation, arXiv:2601.20802, 2026

---

> > ### Author Rebuttal · Reviewer_8img · 2026-04-03
> >
> > Thank the authors for the rebuttal. While concerns regarding experimental scale remain, I understand the time constraints and will maintain my positive score.

---

> > > ### Author Response · Authors · 2026-04-08
> > >
> > > We thank the reviewer for understanding the time constraints and maintaining their positive score following our additional experiments.
> > >
> > > We further extend our evaluation to another **7B** model (Qwen2.5-7B) and a more **diverse multi-task** setting combining **MATH** (Hendrycks et al., 2021), **ARC**, and **SciKnowEval (Chemistry, Physics)**. This setup spans heterogeneous task types, including mathematical reasoning, inductive reasoning (ARC), and natural language QA.
> > >
> > > We train MT-GRPO with λ ∈ {0.05, 0.2} and the primary baselines from the paper (GRPO, DAPO, SEC-GRPO, SEC-DAPO) for 150 steps.
> > >
> > > MT-GRPO **substantially improves** worst-task performance (ARC) and yields more balanced optimization compared to all baselines (**37.6 vs 25.0** for DAPO). Moreover, larger λ improves worst-task accuracy at the cost of average accuracy, consistent with the trade-off discussed in our experimental section (see Figure 7, Section 6.2).
> > >
> > > The results are shown below (see https://anonymous.4open.science/r/icml-rebuttal-8DDE/multi-domain/qwen7B_merged_2x3.png ):
> > >
> > > | Name          | Avg Acc (%) | Worst-Task Acc (%) |
> > > |---------------|------------:|-------------------:|
> > > | SEC_GRPO      | 52.6        | 8.9                |
> > > | GRPO          | 55.4        | 15.0               |
> > > | SEC_DAPO      | 56.1        | 8.8                |
> > > | DAPO          | 58.3        | 25.0               |
> > > | MT-GRPO_0.05   |    56.2     |        33.1        |
> > > | MT-GRPO_0.2   | 51.8        | **37.6**               |
> > >
> > > Runtime analysis further shows that MT-GRPO reaches target worst-task accuracy thresholds **significantly faster** than baselines. (see: https://anonymous.4open.science/r/icml-rebuttal-8DDE/multi-domain/qwen7B_lines_time.png )
> > >
> > > These additional experiments further **strengthen** our findings by demonstrating that MT-GRPO generalizes to larger models and diverse multi-task settings.
> > >
> > > We will include these additional experimental results in the revised version.

---

### Decision · Program_Chairs · 2026-04-30

**Decision:**

Accept (regular)

**Comment:**

The paper addresses optimization imbalance in multi-task RL post-training with GRPO by proposing MT-GRPO, which incorporates improvement-aware reweighting and a ratio-preserving sampler. While initial reviews raised concerns regarding experimental scale (3B models) and task diversity, the authors effectively addressed these during the rebuttal by adding 7B model experiments and diverse benchmarks like SciKnowEval and MATH. Comprehensive ablations and comparisons with sequential training baselines further demonstrated the method's robustness and efficiency. The reviewers are generally satisfied with the technical contributions.